# Endogenous antisense RNA curbs CD39 expression in Crohn's disease

Rasika P. Harshe[1,5], Anyan Xie[2,5], Marta Vuerich[1,5], Luiza Abrahão Frank[1], Barbora Gromova[1,3], Haohai Zhang[1], Rene' J. Robles[2], Samiran Mukherjee[1], Eva Csizmadia[1], Efi Kokkotou[2], Adam S. Cheifetz[2], Alan C. Moss[2], Satya K. Kota[4], Simon C. Robson [1,2,5] & Maria Serena Longhi [1,5✉]

CD39 is an ectonucleotidase that initiates conversion of extracellular nucleotides into immunosuppressive adenosine. CD39 is expressed by regulatory T (Treg)-cells, where it mediates immunosuppression, and by a subset of T-helper (Th) 17-cells, where it limits pathogenicity. CD39 is regulated via single-nucleotide-polymorphisms and upon activation of aryl-hydrocarbon-receptor and oxygen-mediated pathways. Here we report a mechanism of CD39 regulation that relies on the presence of an endogenous antisense RNA, transcribed from the 3'-end of the human *CD39/ENTPD1* gene. CD39-specific antisense is increased in Treg and Th17-cells of Crohn's disease patients over controls. It largely localizes in the cell nucleus and regulates CD39 by interacting with nucleolin and heterogeneous-nuclear-ribonucleoprotein-A1. Antisense silencing results in CD39 upregulation in vitro and amelioration of disease activity in a trinitro-benzene-sulfonic-acid model of colitis in humanized NOD/scid/gamma mice. Inhibition/blockade of antisense might represent a therapeutic strategy to restore CD39 along with immunohomeostasis in Crohn's disease.

---

[1] Department of Anesthesia, Critical Care & Pain Medicine, Beth Israel Deaconess Medical Center, Harvard Medical School, 330 Brookline Avenue, Boston, MA 02215, USA. [2] Division of Gastroenterology, Department of Medicine, Beth Israel Deaconess Medical Center, Harvard Medical School, 330 Brookline Avenue, Boston, MA 02215, USA. [3] Institute of Molecular Biomedicine, Faculty of Medicine, Comenius University, Bratislava, Slovakia. [4] Department of Oral Medicine, Infection and Immunity, Harvard School of Dental Medicine, 188 Longwood Avenue, Boston, MA 02115, USA. [5] These authors contributed equally: Rasika P. Harshe, Anyan Xie, Marta Vuerich, Simon C. Robson, Maria Serena Longhi. ✉email: mlonghi@bidmc.harvard.edu

CD39 regulates immune homeostasis by hydrolyzing adenosine triphosphate (ATP) and adenosine diphosphate into adenosine monophosphate (AMP) that is subsequently converted into adenosine by CD73, the ectoenzyme that works in tandem with CD39[1–3]. CD39 is expressed on various immune cells, including regulatory T (Treg) cells, where it mediates suppressive function[4–6], and by a subset of effector T helper type 17 (Th17) cells, where it marks the acquisition of regulatory properties and limits pathogenic potential[7,8].

We and others have shown that alterations of CD39 expression result in immune dysregulation in inflammatory bowel disease[7–10]. CD39 globally deficient mice are highly susceptible to colitis induced by dextran sulfate sodium[9] and pretreatment with apyrase —a soluble factor with enzymatic activity equivalent to CD39— prevents weight loss in the same model of colitis[9].

CD39 levels and function are decreased in both Treg and Th17 cells derived from the peripheral blood and lamina propria (LP) of Crohn's disease patients[7,8,10], this being associated with defective Treg function and impaired Th17 cell ability to undergo immunoregulation and subsequent perpetuation of pathogenic potential. Conversely, CD39 overexpression confers Treg increased ability to suppress in a T cell transfer model of colitis[11].

CD39 can be regulated at the genetic level through single-nucleotide polymorphisms (SNPs) in noncoding regions of the gene that are associated with low CD39 mRNA expression and with predisposition to the disease[9,12,13]. CD39 is also regulated at the transcriptional level upon activation of aryl hydrocarbon receptor (AhR)[14], a receptor for toxins/xenobiotics that also regulates adaptive immunity[15,16]. Previous studies have shown that unconjugated bilirubin, an endogenous ligand of AhR, confers immunoregulatory properties to Th17 cells, this being dependent upon CD39 induction[8]. Additional control over CD39 expression derives from alterations of oxygen levels[17–20]. We recently found that protracted hypoxia, which is associated with chronic inflammatory statuses, interferes with CD39 levels by inhibiting AhR signaling in Crohn's derived Th17 cells[20].

Additional mechanisms of gene regulation might be associated with the presence of antisense RNAs, a class of long noncoding RNAs that are transcribed from the strand opposite to the sense strand of the overlapping gene. As other long noncoding RNAs, antisense RNAs can be >200 nucleotides; they are poly-A capped and might act through binding DNA, chromatin, RNA, and transcription factors[21].

Antisense RNA plays a role in the posttranscriptional regulation of the genes encoding endothelial nitric oxide synthase, a key enzyme for vascular wall homeostasis[22,23], as well as hypoxia-inducible factor 1-alpha (HIF-1α)[24]. With regard to CD39, inhibition of phosphodiesterase 3, which induces increase in the c-AMP intracellular concentration, results in augmented CD39 protein levels in RAW macrophages[25], suggesting involvement in the posttranslational regulation of CD39. A non-endogenous antisense construct to Epstein–Barr virus LMP1—a gene pivotal to growth transformation and B lymphocyte immortalization— substantively impacts CD39 expression[26], further supporting the role for additional regulatory mechanisms in the control of CD39 gene expression.

Here we report regulation of CD39 by an endogenous antisense RNA transcript, which is present at the 3′ end of the human CD39/ENTPD1 gene within chromosome 10. This antisense RNA is enriched in both Treg and Th17 cells obtained from Crohn's disease patient samples. Mechanistically, it regulates CD39 expression levels upon interactions with nucleolin (NCL) and heterogeneous nuclear ribonucleoprotein A1 (HNRNPA1). Blockade of this antisense RNA using specific oligonucleotides restores CD39 levels in vitro and ameliorates the course of colitis in humanized NOD/scid/gamma mice in vivo.

## Results

**Endogenous antisense RNA at human CD39 locus.** We have previously demonstrated that human CD39 is regulated at the genetic level via SNPs in the promoter region of the gene that are associated with altered CD39 mRNA expression[9] and at the transcriptional level upon engagement of stimulatory or inhibitory pathways governed by AhR and HIF-1α/hypoxia[8,20,27]. Here we aimed to determine whether CD39 could be also regulated via endogenous long noncoding RNAs.

We performed bioinformatic mining of human CD39 locus at 10q24.1. Our search of genome databases identified a predicted long noncoding RNA, with multiple splice variants in antisense orientation to CD39 gene, namely ENTPD1-AS (HGNC:45203), here after referred to as CD39-AS RNA. The longest transcript variant ENTPD1-AS1-230 (ENSG00000226688.7) of CD39-AS RNA spans the entire length of the CD39 gene (Fig. 1a) and does not have coding potential for a protein product. To validate the expression dynamics of CD39-AS RNA in T cells, reverse transcription followed by quantitative polymerase chain reaction (RT-qPCR) was performed on RNA isolated from Jurkat and peripheral blood derived human T cells using different sets of primers spanning distinct regions corresponding to individual splice variants. We identified a primer pair (Supplementary Table 1 and Supplementary Notes) that resulted in reliable amplification of at least two splice variants of CD39-AS RNA, ENTPD1-AS1-209 (ENST00000452728.5), henceforth v1, and the ENTPD1-AS1-201 (ENST00000414006.2), henceforth v2, variants (Fig. 1a). The region amplified by this primer pair corresponds to a 135-base pair and to a 257-base pair-long product, respectively (Supplementary Notes). Both splice variants are located after the 3′ end of, and opposite to, the human CD39/ENTPD1 (henceforth CD39) gene (Fig. 1a). The two variants are 858 and 2457 base pair long and encompass the chromosome 10: 95,875,388–96,090,198 and chromosome 10: 95,873,761–95,907,842 regions.

CD39-AS RNA was readily detected in multiple cell lines of hematopoietic origin including Jurkat, Raji, HCC1739BL, and THP-1 (Fig. 1b). Importantly, CD39 mRNA expression was significantly more and inversely correlated with low CD39-AS RNA levels in HCC1739BL and THP-1 cells (Fig. 1b, c). Jurkat cells showed the highest expression of CD39-AS RNA and relatively low CD39 mRNA levels among these cell lines (Fig. 1b, c). High CD39-AS RNA and low CD39 mRNA expression in Jurkat cells were further confirmed by in situ hybridization using BaseScope and RNAScope analyses (Fig. 1d). Given the high levels of antisense transcripts and the concomitantly low CD39 mRNA expression, Jurkat cells were used as a positive control for all subsequent experiments.

**High CD39-AS levels in Crohn's derived Treg and Th17 cells.** We previously showed reduced CD39 levels and function in Treg and Th17 cells derived from the peripheral blood and LP samples of Crohn's disease patients[7,10]. With the view of defining whether endogenous CD39-AS RNA regulates CD39 levels ex vivo, we tested for CD39-AS RNA expression in Treg and Th17 cells obtained from the peripheral blood and LP biopsied samples of Crohn's disease patients and healthy controls. CD39-AS RNA levels were also measured in Th1 and Th2 cells obtained from a smaller number of healthy controls and patients. Treg and Th17 cells were differentiated from peripheral blood and LP CD4 cells upon exposure to interleukin-2 (IL2), transforming growth factor-β (TGFβ), and Dynabeads CD3/CD28 for T cell expansion for Treg polarization and to IL6, IL1β, and TGFβ for Th17 cell polarization. Th1 and Th2 cells were obtained upon conditioning of peripheral blood CD4 lymphocytes in the presence of IL12 and anti-IL4 (Th1 cells) or IL4 and anti-interferon-γ (anti-IFNγ) (Th2 cells)[8].

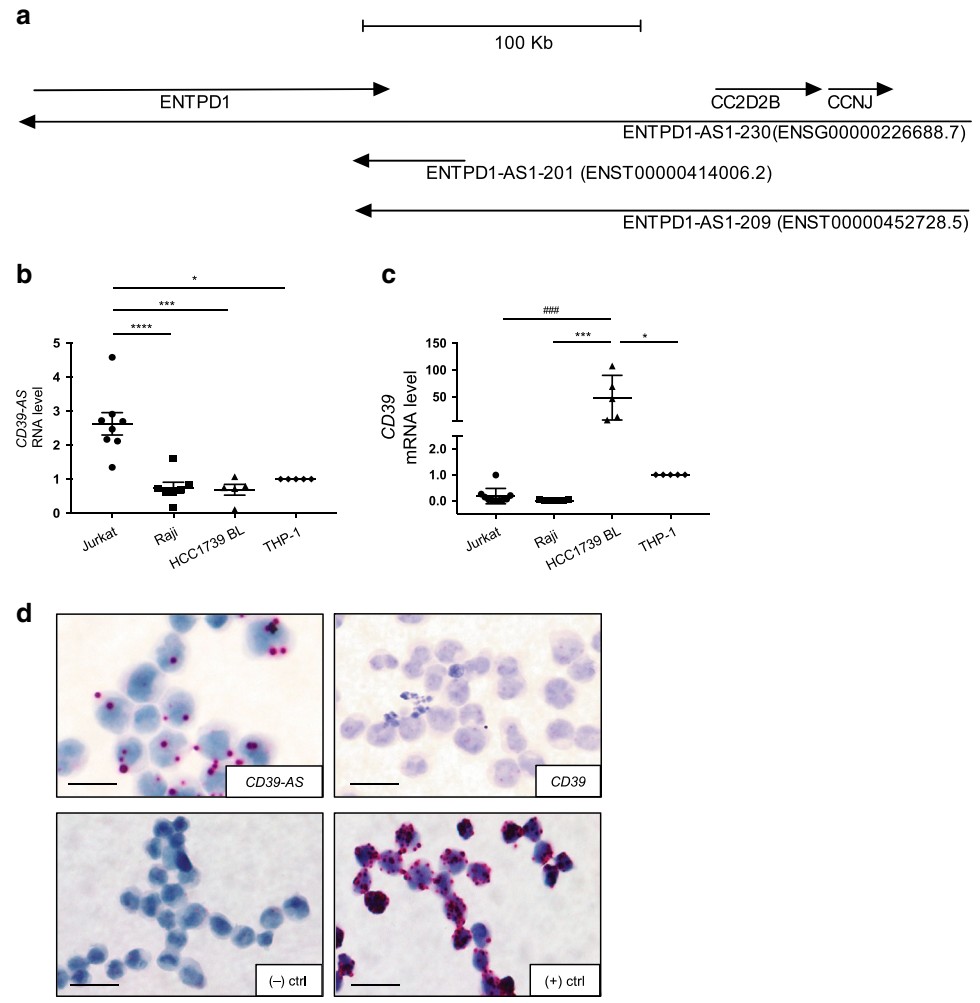

**Fig. 1 CD39-AS RNA is located at the 3′ end of the human CD39 gene and regulates CD39 mRNA levels. a** Graphical representation of the 10q24.1 human locus, including *CD39/ENTPD1*, *CC2D2B*, and *CCNJ* genes, *ENTPD1-AS1-230* (ENSG00000226688.7) long transcript antisense RNA variant, and two *ENTPD1/CD39-AS* splice variants. The two *ENTPD1-AS1-209* (ENST00000452728.5) and *ENTPD1-AS1-201* (ENST00000414006.2) splice variants, predicted by Ensembl and UCSC genome browsers, were experimentally confirmed by qRT-PCR analysis. These were both transcribed from the 3′ end of the human *ENTPD1/CD39* gene. **b, c** Mean ± SEM *CD39-AS* RNA and *CD39* mRNA levels in Jurkat ($n = 8$ replicates), Raji ($n = 7$ replicates), HCC1739BL ($n = 5$ replicates), and THP-1 ($n = 5$ replicates) cell lines. Because of high *CD39-AS* RNA and concomitantly low *CD39* mRNA levels, Jurkat cells were used for all subsequent experiments as positive control for antisense RNA expression. BaseScope and RNAScope chromogenic assays were used to detect *CD39-AS* RNA and *CD39* mRNA transcripts in cytospins of Jurkat cells. **d** Representative images of Jurkat cells stained with *CD39-AS* BaseScope, *CD39* RNAScope, and negative and positive control staining is shown (magnification ×40, scale bar: 25 μM). A representative of three independent experiments is shown. Comparisons in **b, c** were made using one-way ANOVA, followed by Tukey's multiple comparisons test (**b**: *$P = 0.013$; ***$P = 0.0002$; ****$P \leq 0.0001$; **c**: *$P = 0.013$; ***$P = 0.0004$; ###$P = 0.0002$).

Cell phenotype after polarization was verified by flow cytometry (Supplementary Fig. 1a–d). In both healthy controls and Crohn's disease patients, levels of *CD39-AS* RNA and *CD39* mRNA were higher in Treg and Th17 cells, when compared with Th1 and Th2 subsets (Supplementary Fig. 2a, b).

*CD39-AS* RNA was enriched in both Treg and Th17 cells obtained from Crohn's disease patients; whereas *CD39* mRNA levels were significantly decreased in patients when compared to healthy controls (Fig. 2a–d).

*CD39-AS* RNA levels were higher, while *CD39* mRNA levels were lower, in Treg obtained from Crohn's patients who were not under immunosuppressive treatment at the time of study/sample collection (Fig. 2e). Importantly, there was a positive correlation between *CD39-AS* RNA levels with Montreal type in Treg and with Harvey–Bradshaw index (HBI) in Th17 cells (Fig. 2f). When considering Treg and Th17 cells obtained from the LP of Crohn's disease patients, we noted reduced *CD39-AS* RNA levels in Th17

cells derived from non-inflamed as compared to inflamed biopsied areas (Fig. 2g). In the same cells, significant increase in *CD39* mRNA levels was noted (Fig. 2g). Higher *CD39* mRNA levels were also noted in Treg obtained from non-inflamed biopsied areas and this increase trended to significance (Fig. 2g).

**Endogenous CD39-AS RNA regulates the expression of CD39.** In order to define to what extent endogenous antisense RNA mediates the regulation of CD39 expression, we inhibited *CD39-AS* RNA function using specific self-delivering FANA CD39 antisense (FANA-CD39-AS) oligonucleotides. Two FANA-CD39-AS oligos targeting specific regions within the v1 or v2 variant of *CD39-AS* RNA (Supplementary Notes) were used to specific silencing. Addition of FANA-CD39-AS oligonucleotides to Jurkat cells progressively decreased *CD39-AS* RNA over a 72-h period and increased *CD39* mRNA levels (Fig. 3a, b).

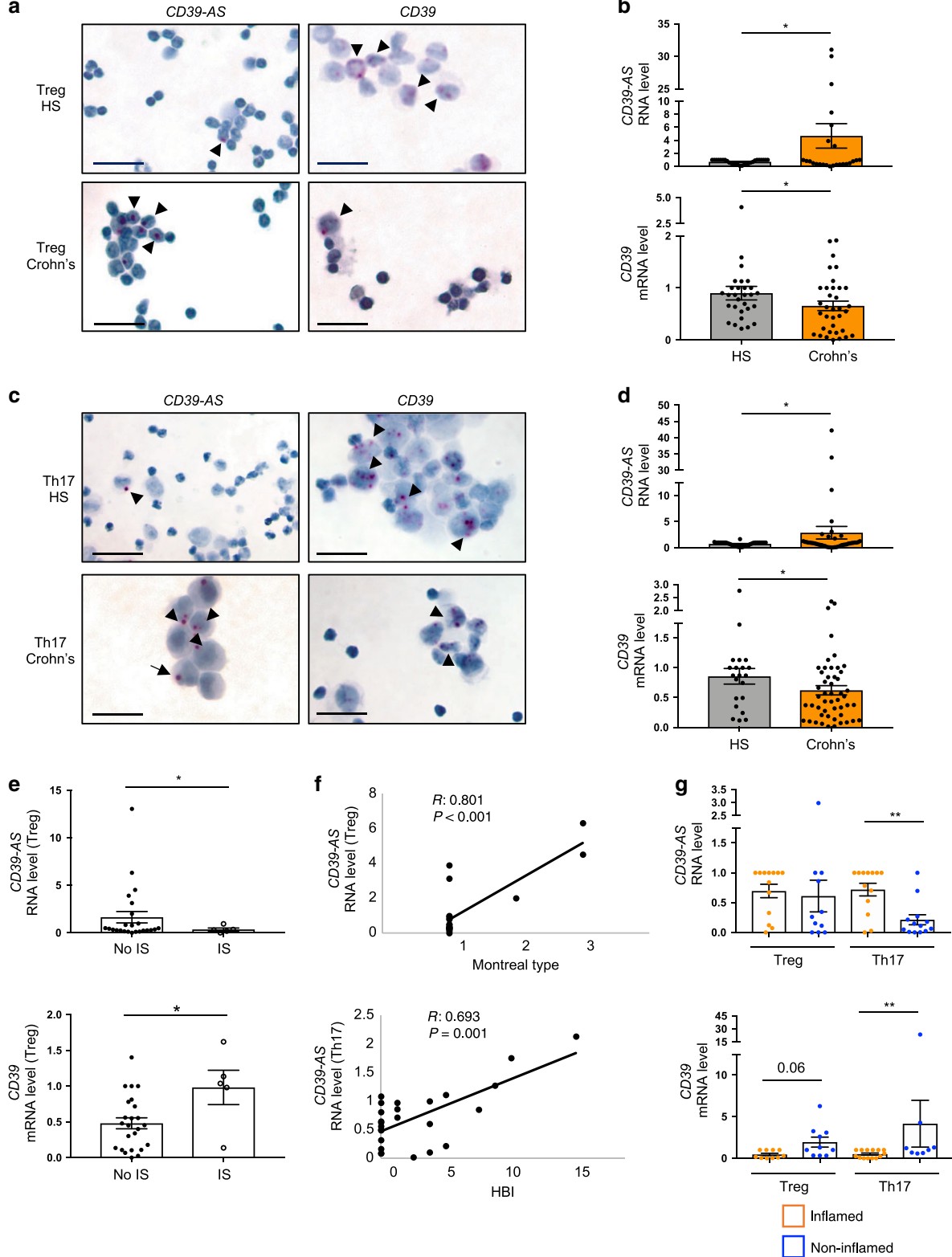

Concomitant with increase in mRNA levels, higher frequency of CD39$^+$ cells and CD39 mean fluorescence intensity (MFI) were also found upon *CD39-AS* RNA silencing in the presence of FANA-CD39-AS oligonucleotides (Fig. 3c and Supplementary Fig. 3a). Similar to Jurkat cells, in Treg and Th17 cells from healthy controls incubation with FANA-CD39-AS oligonucleotides resulted in decreased *CD39-AS* RNA levels, increased *CD39* mRNA, higher frequency of CD39$^+$ cells, and heightened CD39

MFI (Supplementary Figs. 3b and 4a–c). Based on these data indicating that the effects of FANA-CD39-AS oligonucleotides on Jurkat cells and healthy control Treg and Th17 cells were mainly evident at 72 h, further experiments were conducted to test the effects of FANA-CD39-AS addition in Treg and Th17 cells obtained from a larger number of healthy controls and Crohn's disease patients (Fig. 3d, e). As noted in healthy subjects, exposure of Treg and Th17 cells derived from Crohn's disease patients,

**Fig. 2 *CD39-AS* RNA levels are increased in Crohn's derived Treg and Th17 cells.** Treg and Th17 cells were derived upon polarization of peripheral blood CD4 cells of healthy blood donors and Crohn's disease patients. Presence of *CD39-AS* RNA and *CD39* mRNA transcripts was detected in cytospins of differentiated Treg and Th17 cells using BaseScope and RNAScope chromogenic assays. **a** Representative images of *CD39-AS* RNA and *CD39* mRNA detection in cytospins of Treg derived from one healthy blood donor and one patient with Crohn's disease (magnification ×40, scale bar: 25 μM) are shown. **b** Mean ± SEM *CD39-AS* RNA and *CD39* mRNA levels in Treg from healthy subjects (HS, $n = 27$ for *CD39-AS* and $n = 29$ for *CD39*) and Crohn's disease patients ($n = 25$ for *CD39-AS* and $n = 35$ for *CD39*) (*$P = 0.03$ for *CD39-AS* RNA and *$P = 0.013$ for *CD39* mRNA using two-sided unpaired $t$ test). **c** Representative images of *CD39-AS* RNA and *CD39* mRNA detection in cytospins of Th17 cells derived from one healthy blood donor and one patient with Crohn's disease (magnification ×40, scale bar: 25 μM). **d** Mean ± SEM *CD39-AS* RNA and *CD39* mRNA levels in Th17 cells from healthy subjects (HS, $n = 38$ for *CD39-AS* and $n = 21$ for *CD39*) and Crohn's disease patients ($n = 44$ for *CD39-AS* and $n = 50$ for *CD39*) (*$P = 0.048$ for *CD39-AS* RNA and *$P = 0.042$ for *CD39* mRNA using two-sided unpaired $t$ test). **e** Mean ± SEM *CD39-AS* RNA and *CD39* mRNA levels in Treg of Crohn's disease patients treated without ($n = 24$) or with ($n = 5$) immunosuppressive (IS) drugs (*$P = 0.047$ for *CD39-AS* RNA and *$P = 0.017$ for *CD39* mRNA using two-sided unpaired $t$ test). **f** Correlation between *CD39-AS* RNA levels with Montreal type in Treg and with Harvey–Bradshaw Index in Th17 cells (correlation made using Pearson correlation coefficient). **g** Mean ± SEM *CD39-AS* RNA and *CD39* mRNA levels in Treg and Th17 cells obtained from lamina propria mononuclear cells of Crohn's disease patients. *CD39-AS* RNA and *CD39* mRNA levels in cells from inflamed (*CD39-AS*: Treg and Th17, $n = 13$; *CD39*: Treg, $n = 10$; Th17, $n = 14$) and non-inflamed biopsied areas (*CD39-AS*: Treg, $n = 11$; Th17, $n = 13$; *CD39*: Treg, $n = 10$; Th17, $n = 8$) (**$P = 0.002$ for *CD39-AS* RNA and **$P = 0.006$ for *CD39* mRNA using two-sided unpaired $t$ test).

resulted in decreased *CD39-AS* RNA levels (Fig. 3d), higher *CD39* mRNA levels (Fig. 3d), increased frequencies of CD39⁺ cells (Supplementary Fig. 3c), and heightened CD39 MFI (Fig. 3e).

No significant changes were noted in the levels of *CD39-AS* RNA, *CD39* mRNA, and CD39 MFI following exposure of Th1 and Th2 cells to FANA-CD39-AS oligonucleotides in healthy subjects and Crohn's disease patients (Supplementary Fig. 5a–d). IFNγ and IL4 MFIs were also unchanged after Th1 or Th2 exposure to FANA-CD39-AS oligonucleotides (Supplementary Fig. 5c, d).

*CD39-AS* RNA silencing left unchanged FOXP3 MFI in Jurkat, Treg, and Th17 cells over the 72-h culture period (Supplementary Fig. 6a, b).

Importantly, exposure to FANA-CD39-AS oligonucleotides resulted in improved Treg-suppressive function in Crohn's disease patient samples and reverted the effects of CD39 silencing on Treg function in both healthy subjects and patients (Fig. 3f).

Together, these results show that *CD39-AS* RNA regulates *CD39* mRNA levels and MFI in both Treg and Th17 cells from healthy controls as well as patient samples. Silencing of *CD39-AS* RNA boosts the suppressive function of Treg from Crohn's disease patients.

**Blockade of CD39-AS RNA ameliorates experimental colitis.** Increased expression of CD39 upon antisense silencing in T cell subsets derived from Crohn's disease patients led us to investigate potential therapeutic effects of FANA-CD39-AS oligonucleotides in vivo in a well-characterized mouse model of T cell-mediated colitis[28]. NOD/scid/gamma recipient mice were injected with human CD4⁺ cells from healthy donors (Fig. 4a) that showed *CD39-AS* RNA expression (Fig. 4b and Supplementary Fig. 7a, b). Mice were checked for CD4 T cell reconstitution 3 weeks after injection and were subjected to trinitrobenzene sulfonic acid (TNBS) sensitization (Fig. 4a), as previously described[28]. Seven days later, mice were exposed to TNBS and at the same time treated with vehicle or FANA-CD39-AS oligonucleotides and sacrificed 72 h later. We found that the disease activity index was higher in vehicle than in FANA-CD39-AS oligonucleotide-treated animals (Fig. 4c) that also displayed greater colon length (Fig. 4d) and lower histology score (Fig. 4e) at the time of harvest. Flow cytometric analysis of CD4 cells derived from the peripheral blood, spleen, mesenteric lymph nodes (MLNs), intraepithelial lymphocyte (IEL), and LP lymphocyte compartments showed that in mice treated with FANA-CD39-AS oligonucleotides there was an increase in the frequency of CD4⁺CD39⁺ cells in the peripheral blood (Fig. 4f, g and Supplementary Fig. 8a, b) and a

decrease in the proportion of IL17-producing CD4 cells in the peripheral blood, spleen, and—albeit not significantly—IELs (Fig. 4g and Supplementary Fig. 8d). There were no significant differences in the frequency of CD4⁺FOXP3⁺ and CD4⁺IL10⁺ lymphocytes (Fig. 4g and Supplementary Fig. 8c, e) or in the proportion of CD4 lymphocytes expressing CD25, RORC, and IFNγ between vehicle and FANA-CD39-AS-treated mice in all the compartments studied. Treatment with FANA-CD39-AS resulted in heightened CD39 MFI in CD4⁺FOXP3⁺, CD4⁺IL17⁺, and CD4⁺IL10⁺ cells obtained from the peripheral blood (Supplementary Fig. 8f).

Overall, these data show that mice exposed to TNBS colitis after human CD4 T cell replenishment and concomitantly treated with FANA-CD39-AS oligonucleotides display a more benign course of colitis while showing a more immunoregulatory phenotype in peripheral blood and spleen and—to a lesser extent—IEL-derived CD4 cells.

To determine whether FANA-CD39-AS oligonucleotides have also an effect on Treg function in vivo, we reconstituted NOD/scid/gamma mice with CD4⁺ cells obtained from one healthy blood donor, and at the time of TNBS administration, we injected these recipients with FANA-CD39-AS-treated CD4⁺CD25^highCD127^low Treg, immunomagnetically sorted from the peripheral blood of the same donor (Supplementary Fig. 9a). Expression of *CD39* mRNA and *CD39-AS* RNA of Treg prior to antisense silencing is shown in Supplementary Fig. 9b. At the time of harvest, mice injected with FANA-CD39-AS-treated Treg displayed the lowest disease activity index (Supplementary Fig. 9c), the highest colon length (Supplementary Fig. 9d), and the lowest histology score (Supplementary Fig. 9e), when compared to recipients administered untreated Treg or vehicle.

**CD39-AS interacts with NCL and HNRNPA1.** In order to understand how antisense might regulate CD39 levels, we first evaluated the cellular distribution of *CD39-AS* RNA in the nuclear and cytosolic RNA fractions of Jurkat cells. Validation of subcellular fractioning is presented in Supplementary Fig. 10a, b. *CD39-AS* RNA was predominantly localized in the nuclear compared to cytosolic RNA fraction (Fig. 5a).

Next, to identify nuclear proteins that interact with *CD39-AS* RNA and potentially aid in regulation of *CD39* RNA levels, we performed RNA pulldown assay on nuclear extracts from Jurkat cells. The two highly expressed *CD39-AS* splice variants—v1 and v2—were in vitro transcribed and purified RNA was incubated with nuclear extracts. *CD39-AS* RNA-bound proteins were

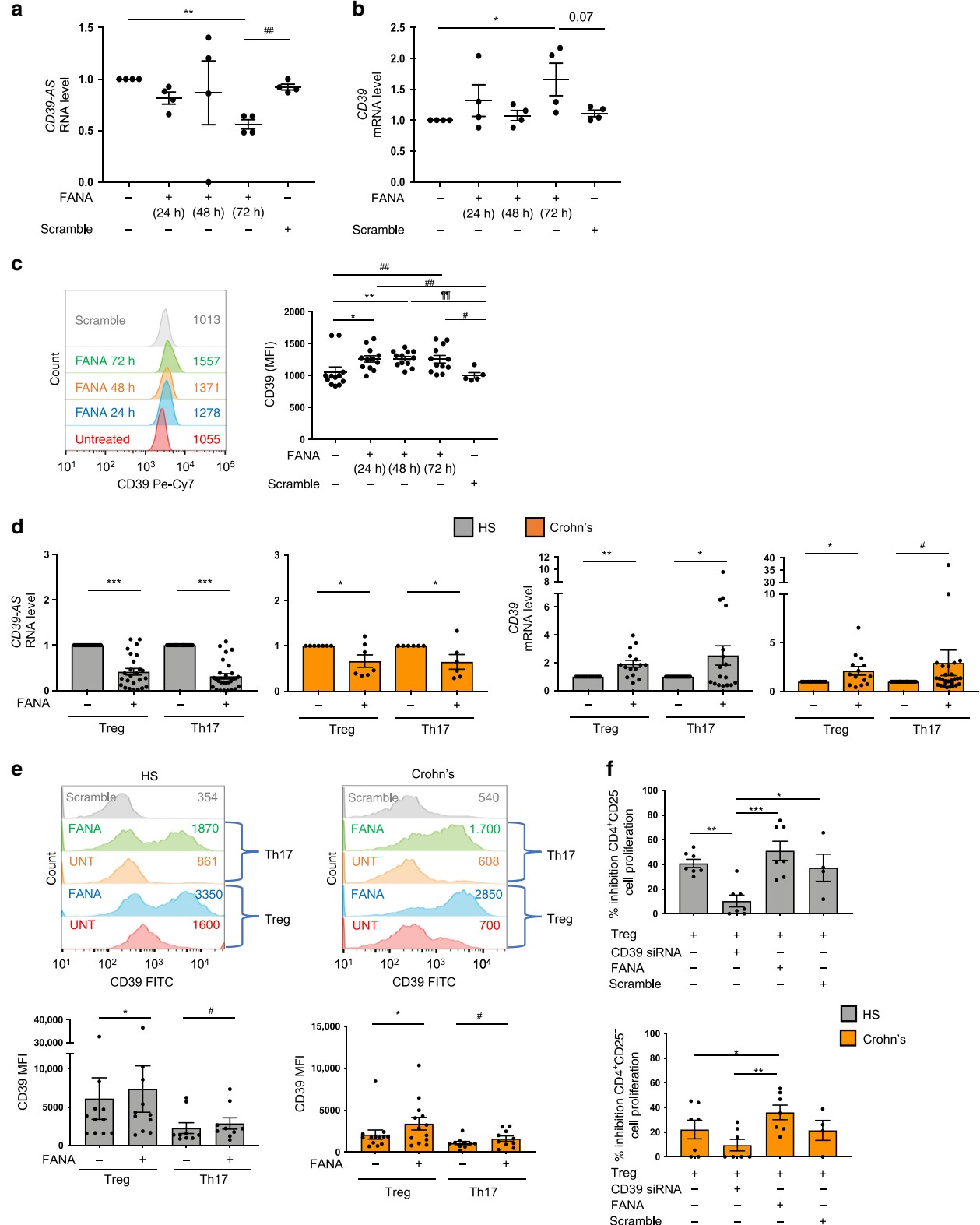

affinity purified and protein identification was carried out via mass spectrometry thereafter. Of the several nuclear proteins that were identified in the bound fraction for both splice variants, NCL and HNRNPA1 were the most abundant (Fig. 5b and Supplementary Tables 2 and 3). To interpret the functional relevance of interaction between *CD39-AS* RNA and NCL and HNRNPA1 proteins, we silenced the cellular levels of NCL and HNRNPA1 using specific small interfering RNA (siRNA;

Supplementary Fig. 11a, b). Silencing of both NCL and HNRNPA1 resulted in significant upregulation of *CD39* mRNA levels (Fig. 5c) and increased CD39 MFI (Fig. 5d).

Subcellular localization analyses of Treg and Th17 cells indicated that, akin to Jurkat cells (Fig. 5a), *CD39-AS* RNA levels were much more abundant in the nucleus of both cell types in healthy subjects and Crohn's disease patients (Fig. 6a, b). siRNA silencing of NCL or HNRNPA1, alone or together, increased

**Fig. 3 Inhibition of antisense RNA boosts *CD39* mRNA levels and MFI.** Jurkat cells were exposed to FANA-CD39-AS oligonucleotides (or scramble) specifically inhibiting the *ENST00000452728.5* (v1) or the *ENST00000414006.2* (v2) splice variant. **a, b** *CD39-AS* RNA and *CD39* mRNA expression was measured (*n* = 4 replicates). Results obtained for FANA-CD39-AS oligonucleotides specific to individual variants are pooled (**a**: **P = 0.002, ##P = 0.001; **b**: *P = 0.046 using one-way ANOVA followed by Tukey's multiple comparisons test). **c** CD39 expression in the absence and presence of FANA-CD39-AS oligonucleotides was also assessed by flow cytometry and expressed as MFI. A representative image is shown. MFI values are indicated within the histogram. Mean ± SEM CD39 MFI is also shown (*n* = 12 replicates per condition; *P = 0.018, **P = 0.009, #P = 0.02, ##P = 0.003, ¶¶P = 0.001 using one-way ANOVA followed by Tukey's multiple comparisons test). In subsequent experiments, FANA-CD39-AS oligonucleotides were added to Treg and Th17 cells for 72 h. Results obtained in the presence of FANA-CD39-AS oligonucleotides specific to individual variants are pooled. **d** Mean ± SEM *CD39-AS* RNA and *CD39* mRNA levels in healthy subjects (HS, *CD39-AS*, Treg: *n* = 25; Th17, *n* = 28; *CD39*, Treg: *n* = 15; Th17, *n* = 17) and Crohn's disease patients (*CD39-AS*, Treg: *n* = 7; Th17, *n* = 6; *CD39*; Treg: *n* = 14; Th17, *n* = 28) (***P < 0.001, *P = 0.03 for *CD39-AS* RNA; **P = 0.001, *P = 0.035 for *CD39* mRNA in HS; *P = 0.017, #P = 0.049 for *CD39* mRNA in Crohn's patients using two-sided paired *t* test). **e** A representative histogram of CD39 MFI in Treg and Th17 cells from one HS and one Crohn's patient is shown. Values of CD39 MFI are indicated within the histogram. Mean ± SEM CD39 MFI in Treg and Th17 cells from HS (Treg, *n* = 11; Th17, *n* = 9) and Crohn's disease patients (Treg, *n* = 13; Th17, *n* = 10) is shown (*P = 0.048, #P = 0.016 for comparisons in HS; *P = 0.02, #P = 0.039 for comparisons in Crohn's patients using two-sided paired *t* test). **f** Treg suppression of CD4+CD25− cell proliferation was assessed in a co-culture experiment using ³H-thymidine incorporation. Mean ± SEM percentage inhibition of CD4+CD25− cell proliferation in the presence of untreated, CD39 siRNA, FANA, or scramble-treated Treg is shown (HS *n* = 7; Crohn's disease patients *n* = 7) (*P = 0.03, **P = 0.009, ***P = 0.0006 for comparisons in HS; *P = 0.04, **P = 0.004 for comparisons in Crohn's patients using one-way ANOVA, followed by Tukey's multiple comparisons test).

*CD39* mRNA levels and MFI in Treg (Fig. 6c) and Th17 cells (Fig. 6d) of Crohn's disease patients. No significant changes were observed in *CD39* mRNA levels and MFI following NCL and HNRNPA1 silencing in Treg and Th17 cells obtained from healthy subjects (Fig. 6c, d) and in the levels of *CD39-AS* RNA in Treg and Th17 cells from both Crohn's disease patients and healthy subjects (Fig. 6c, d).

Importantly, silencing of NCL, HNRNPA1, or both boosted the suppressive function of Treg derived from Crohn's disease patients (Fig. 6e); such an effect was not noted in Treg derived from healthy controls.

These findings indicate that antisense regulates *CD39* mRNA levels and MFI upon interaction with NCL and HNRNPA1. Silencing of these two proteins boost the CD39 levels of expression and Treg-suppressive function in Crohn's derived cells.

## Discussion

In this paper, we have identified and characterized an antisense long noncoding RNA at CD39 locus. *CD39-AS* RNA has no coding potential and controls *CD39* gene expression by regulating the mRNA levels. *CD39-AS* RNA is enriched in Treg and Th17 cells derived from Crohn's disease patients where their expression levels are also associated with markers of disease activity. Importantly, and of high translational and clinical relevance, treatment with oligonucleotides inhibiting the *CD39-AS* RNA ameliorates the course of TNBS colitis in NOD/scid/gamma mice, adoptively transferred with antisense+ human CD4 cells.

Our data provide evidence that endogenous antisense RNA can regulate *CD39* gene expression. This mode of gene regulation has possible clinical implications given the role of CD39 in governing immunoregulatory cell function and, more in general, immune homeostasis, by hydrolyzing inflammatory ATP. It is known that ATP mediates inflammation and favors the differentiation of Th17 cells in the LP[29]; mechanisms that contribute to the regulation of and promote systemic or local expression of CD39 would aid contrasting the detrimental effects of ATP, whether produced by commensal gut bacteria or resulting from the host ongoing inflammation.

Multiple modes of *CD39* gene regulation have been described thus far in addition to the long noncoding RNA-mediated control of *CD39* mRNA reported here. Previously described mechanisms of CD39 regulation include SNPs in the noncoding region of *CD39/ENTPD1*[9] and alterations in the expression and/or function of AhR and other related factors[8,14,20,27]. Whether *CD39* gene

regulatory mechanisms operating at the genetic, transcriptional, and posttranscriptional levels act in concert within a certain cell type or intervene individually at different time points is still unclear. Our previous data indicate that HIF-1α upregulation during chronic, protracted inflammation—as in the case of Crohn's disease—renders Th17 cells refractory to AhR-mediated immunoregulation, suggesting that CD39 regulation at the transcriptional level is subjected to a certain degree of modulation that might depend on inflammatory statuses as well as environmental triggers[20]. That different mechanisms of CD39 regulation are present and might depend on specific cell types is also supported by our initial results in cell lines, where a clear correlation between *CD39* mRNA and *CD39-AS* RNA levels is present in Jurkat and HCC1739BL but not in Raji or THP-1 cell lines.

Endogenous *CD39-AS* RNA is present in both Treg and Th17 cells at baseline conditions, implicating that such level of regulation could be cell intrinsic. Low levels of *CD39-AS* RNA in other T cell subsets like Th1 and Th2 lymphocytes suggest that this mechanism of CD39 regulation does not operate in all cell types and could be specific to certain T cell subsets.

The evidence that increased expression levels of *CD39-AS* RNA seen in cells derived from patients with Crohn's disease also correlate with the extent of disease activity indicates its modulation during inflammation, similar to other factors contributing to CD39 transcriptional regulation[8,20]. Whether factors modulating CD39 at the transcriptional level, like changes in $O_2$ concentration and activation of hydrocarbon pathways, are also involved in the induction or regulation of *CD39-AS* RNA awaits further investigations.

Inhibition studies using FANA-CD39-AS oligonucleotides might have important implications for the treatment of Crohn's disease. This type of intervention based on antisense RNA silencing aims at restoring CD39 levels and, consequently, immune homeostasis by targeting endogenous antisense variants present in individual cell types. Importantly, in vivo treatment with FANA-CD39-AS oligonucleotides results in a more favorable course of TNBS colitis induced in immunodeficient NOD/scid/gamma mice, previously re-populated using human CD4 cells with *CD39-AS* RNA expression. The clinical phenotype is accompanied by upregulation of CD39 expression in CD4+FOXP3+, CD4+IL10+, and CD4+IL17+ subsets in the peripheral blood and by a concomitant decrease in IL17-producing CD4 cells in peripheral blood, spleen, and—although to a lesser extent—IEL compartment. These findings indicate that blockade of antisense leads to improvement of disease activity and to

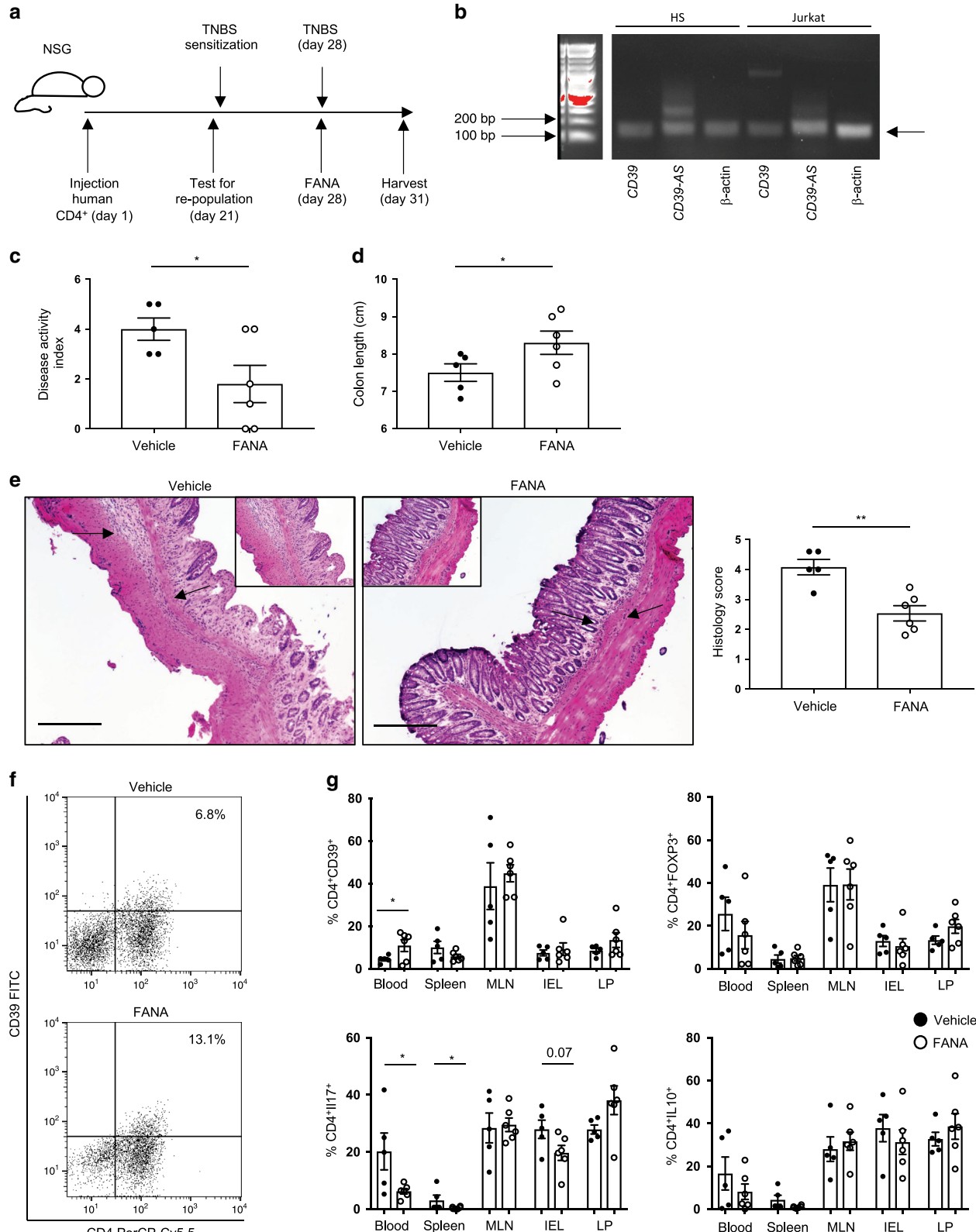

correction of pro-inflammatory phenotypes likely by boosting CD39 levels. The less marked effect of FANA treatment on the phenotype of colonic CD4 cells suggests that the regulation of cell phenotypes within this compartment might result from a combination of factors that also include cytokine milieu, interactions with other cell types, and presence of additional regulatory circuits that might operate at the local level. Whether the increase in CD39 MFI within circulating $CD4^+FOXP3^+$, $CD4^+IL17^+$, and $CD4^+IL10^+$ lymphocytes also results in the regulation of inflammatory cell trafficking, as previously reported[30], requires further investigations.

Our data also indicate that, at the concentration used, single treatment of FANA-CD39-AS oligonucleotides inhibits $CD39\text{-}AS$ RNA while boosting CD39 expression for up to 72 h. Further,

**Fig. 4 Antisense RNA silencing ameliorates TNBS-induced experimental colitis in humanized NOD/scid/gamma mice. a** NOD/scid/gamma (NSG) female recipients were injected with antisense+ CD4 cells, obtained from one healthy blood donor. After 3 weeks, mice were bled and checked for human chimerism. Mice showing >10% human chimerism were sensitized to TNBS. One week after sensitization, mice were administered a single enema of TNBS and given a single injection of FANA-CD39-AS oligonucleotides (or vehicle) intraperitoneally. After 72 h, mice were sacrificed. **b** Cropped gel of RT-PCR showing positivity for *CD39-AS* RNA in healthy blood donor CD4 T lymphocytes (representative of three independent experiments with similar results). Full scan gel is provided in Source data file. **c** Mean ± SEM disease activity index at the time of harvest in TNBS mice treated with vehicle ($n = 5$) or FANA-CD39-AS oligonucleotides ($n = 6$; *$P = 0.035$ using two-sided unpaired $t$ test). **d** Mean ± SEM colon length (cm) at the harvest in vehicle ($n = 5$) and FANA-CD39-AS-treated ($n = 6$) mice (*$P = 0.04$ using two-sided Mann–Whitney test). **e** Hematoxylin and eosin staining of colon sections (original magnification, ×10, scale bar: 200 μM); arrows indicate the area magnified in the insets (×20, scale bar: 200 μM); mean ± SEM histology score at the time of harvesting is also shown (**$P = 0.002$ using two-sided unpaired $t$ test). **f** Representative dot plots of CD4 and CD39 fluorescence of peripheral blood mononuclear cells isolated from one vehicle and one FANA-CD39-AS oligonucleotide-treated mouse at the time of harvest. Frequency of CD4+CD39+ cells is indicated in the upper right quadrant. **g** Mean ± SEM frequency of CD4+CD39+, CD4+FOXP3+, CD4+IL17+, and CD4+IL10+ cells in the peripheral blood, spleen, mesenteric lymph node (MLN), intra-epithelial lymphocytes (IEL), and lamina propria (LP) lymphocytes at the time of harvest in vehicle ($n = 5$) or FANA-CD39-AS oligonucleotide ($n = 6$) treated animals (*$P = 0.048$ for CD4+CD39+, *$P = 0.049$ for CD4+IL-17+ using two-sided unpaired $t$ test).

addition of FANA-CD39-AS oligonucleotides to Treg boosts the suppressive function of these cells in vitro in samples from Crohn's disease patients, where high levels of *CD39-AS* RNA are present, and in vivo, where injection of FANA-CD39-AS oligonucleotide-treated human Treg results in beneficial effects in a model of TNBS colitis in NOD/scid/gamma recipients reconstituted with human CD4+ lymphocytes.

That silencing of *CD39-AS* RNA does not have effect on FOXP3 expression—as supported by our in vitro and in vivo data—indicates the lack of a clear interaction between this antisense RNA and FOXP3 and further emphasizes the specificity of this mechanism of regulation for CD39.

RNA pulldown and mass spectrometry have revealed NCL and HNRNPA1 as two prominent RNA-binding proteins that associate with CD39-AS and whose silencing in T cells correlates with increased CD39 expression. NCL is a highly expressed nucleolar protein that can also shuttle between the nucleus and the cytoplasm and binds to the T cell receptor complex in T cells[31] and granzyme B in cytotoxic T lymphocytes (CTLs)[32], being therefore involved in the process of T cell activation and apoptosis consequent to the CTL-mediated lysis of target cells. HNRNPA1 is also a nucleocytoplasmic shuttling protein with a role in stress response and T cell activation[33,34]. The evidence that siRNA-mediated silencing of NCL and HNRNPA1 boosts CD39 levels in Treg and Th17 cells suggests that *CD39-AS* RNA interactions with both these proteins regulates CD39 by impacting cell activation. Consistent with this, silencing of NCL and HNRNPA1 resulted in upregulation of CD39 only in cells that expressed high levels of *CD39-AS* RNA like Treg and Th17 cells derived from Crohn's disease patients but not from healthy subjects. These findings are corroborated by the evidence of increased Treg-suppressive function following silencing of NCL, HNRNPA1, or both in Crohn's disease but not in healthy control-derived cells.

This suggests that there might be intrinsic differences in healthy versus disease-associated cells in the formation of *CD39-AS* RNA–protein complexes and these might be possibly due to relatively low baseline levels of antisense RNA in cells from healthy individuals; therefore, in this context it is plausible that no significant effect is achieved upon silencing of the proteins mainly interacting with the antisense.

Further studies should be directed at defining the exact mechanistic insights that result in CD39 inhibition after the formation of CD39-AS/NCL and/or HNRNPA1 binding. Whether binding to NCL and HNRNPA1 represents the only mechanism through which antisense regulates CD39 is unknown. The longest transcript variant of CD39-AS spans the entire length of the *CD39* gene, therefore implicating the possibility of translational inhibition by direct interaction between the antisense RNA with *CD39* mRNA. Future investigations should be also aimed at evaluating this possibility.

In conclusion, we have found an endogenous antisense long noncoding RNA that regulates *CD39* mRNA level and MFI, being enriched in the nucleus of Crohn's derived Treg and Th17 cells. Silencing of this antisense RNA restores CD39 levels in vitro along with enhancing Treg-suppressive function in patients' samples. Of high relevance for Crohn's disease treatment, *CD39-AS* RNA silencing ameliorates the course of experimental colitis in vivo. This will aid devising innovative therapeutic strategies based on the inhibition of CD39-AS to re-establish immunotolerance in Crohn's disease as well as other chronic inflammatory conditions associated with CD39 defects.

## Methods

**Cell lines.** Jurkat, Raji, THP-1, and HCC1739BL human cells were obtained from American Type Culture Collection (ATCC, Manassas, VA) and maintained at 37 °C and 5% $CO_2$ in RPMI 1640 medium supplemented with 2 mM L-glutamine, 100 IU/ml penicillin, 100 mg/ml streptomycin, 1% non-essential amino acids, and 10% fetal bovine serum (FBS; Thermo Fisher Scientific, Waltham, MA).

**Subjects.** Peripheral blood mononuclear cells (PBMCs) and lamina propria mononuclear cells (LPMCs) were isolated from 70 patients with Crohn's disease (median HBI 3, range 1–12), recruited from the Gastroenterology Division, Beth Israel Deaconess Medical Center (BIDMC), Boston, MA. Thirty patients were studied during active disease, while the remaining were on clinical remission. At the time of the study, 31 patients were on infliximab, 16 were on steroids, and 8 were receiving mercaptopurine. Demographic and clinical data of patients with Crohn's disease are summarized in Supplementary Table 4. PBMCs were also obtained from 38 healthy blood donors (Blood Donor Center at Children's Hospital, Boston, MA). Human studies received Institutional Review Board approval (protocol # 2011P000202) by the Committee on Clinical Investigations, BIDMC. Written informed consent was obtained from all study participants or legally authorized representatives prior to inclusion in the study.

**CD4 cell isolation and T cell polarization.** PBMCs were obtained by density gradient centrifugation on Ficoll-Paque (GE Healthcare Life Sciences, Pittsburgh, PA)[7,8]. LPMCs were obtained from freshly biopsied colonic tissue of 27 patients with Crohn's disease. In 6 of these patients, tissue was collected from both inflamed and non-inflamed areas (3–4 samples per biopsied area). PBMC and LPMC cell viability always exceeded 98%. Treg and Th17 cells were polarized from total CD4 cells, purified from PBMC and LPMC preparations according to the manufacturer's recommendations (Miltenyi Biotec, San Diego, CA). The purity of the sorted CD4+ cells exceeded 92%. After purification, CD4+ cells were resuspended in RPMI 1640 supplemented with 10% FBS and exposed to Treg, Th17, Th1, or Th2 polarizing conditions. These consisted of IL2 (100 ng/ml, R&D Systems, Minneapolis, MN), TGFβ (10 ng/ml, R&D Systems), and Dynabeads Human T activator CD3/CD28 for T cell expansion (bead/cell ratio: 1:2, Thermo Fisher Scientific, Cambridge, MA) for Treg; IL6 (50 ng/ml, R&D Systems), TGFβ (3 ng/ml), IL1β (25 ng/ml, R&D Systems), and Dynabeads Human T activator CD3/CD28 at 1:50 for Th17 cells; IL12 (20 ng/ml, Peprotech, Rocky Hill, NJ), and anti-human IL4 antibodies (10 μg/ml, R&D Systems) for Th1 cells; and IL4 (10 ng/ml, Peprotech) and anti-human IFNγ antibodies (10 μg/ml, R&D Systems) for Th2 cells.

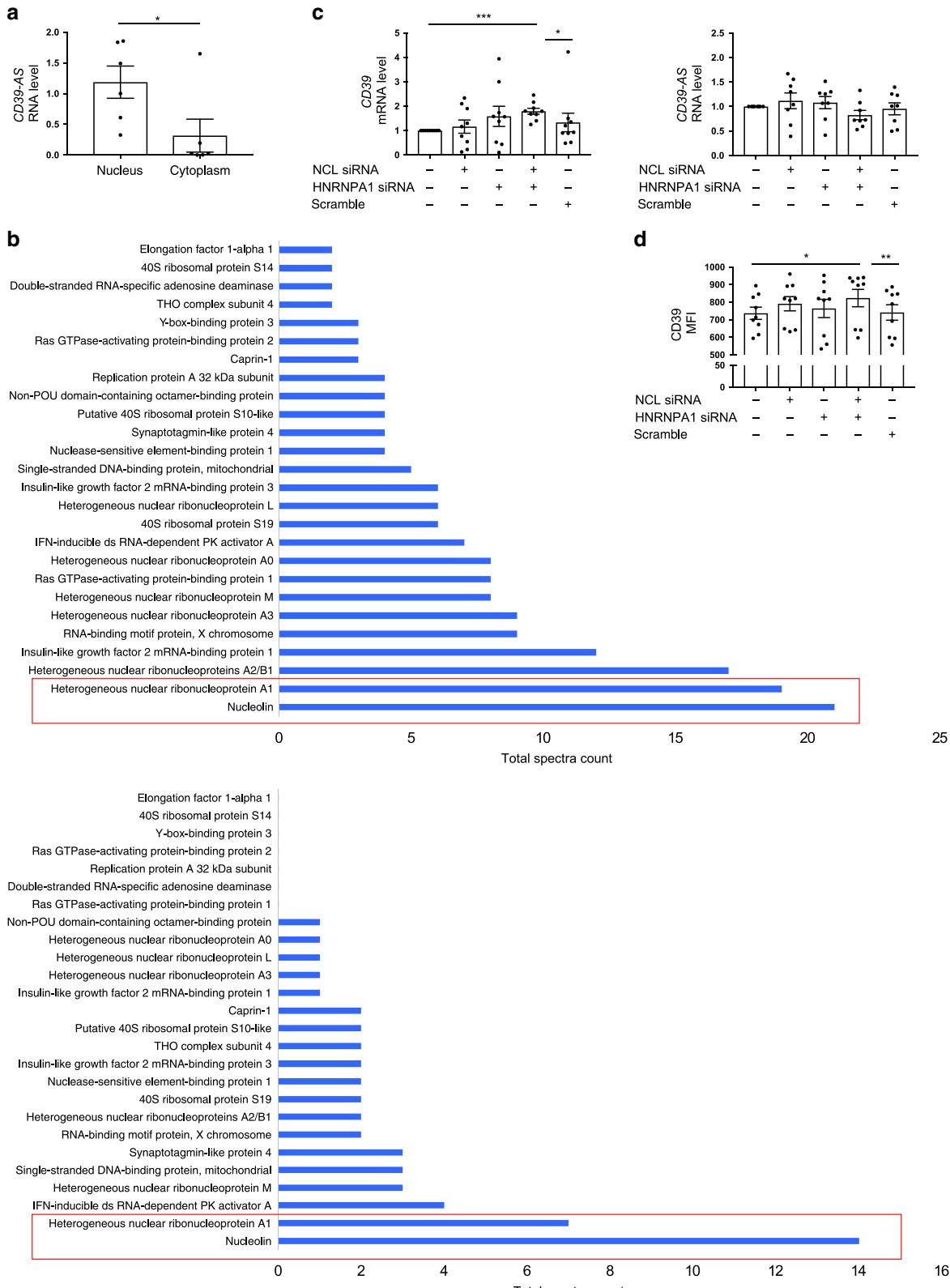

**Flow cytometry**. Treg and Th17 cell phenotype was verified using flow cytometry. Cells were stained with fluorescein isothiocyanate (FITC), Pacific Blue, phycoery-thrin (PE), allophycocyanin (APC)-Cy7, PE-Cy7, APC, or Alexa-Fluor 647-con-jugated monoclonal antibodies to human CD4 (clone # OKT4), CD25 (clone # M-A251), CD127 (clone # A019D5), CD39 (clone # A1), and CCR6 (clone # G034E3), all from Biolegend, San Diego, CA, and to IL23 receptor (IL23R, clone # 218213) from R&D Systems, Minneapolis, MN. Expression of FOXP3 and RORC was assessed by intracellular staining, following cell fixation and permeabilization with Cytofix/Cytoperm (BD Biosciences, San Jose', CA) and incubation with PE- and APC-conjugated antibodies to human FOXP3 (clone # PCH101) and RORC (clone # AFKJS-9), all from eBioscience, San Diego, CA. Frequency of IFNγ- and IL4-producing cells by Th1 and Th2 lymphocytes was determined after exposure to cell stimulation cocktail plus protein transport inhibitors (eBioscience) at 2 μl/ml according to the manufacturer's instructions for 5 h. Staining was carried out using APC anti-human IFNγ (clone # 4 S.B3, Biolegend) and APC anti-human IL4 (clone # 8D4-8, Biolegend). Cells were acquired on a BD LSRII (BD Biosciences) and

**Fig. 5 *CD39-AS* RNA is predominantly localized in the nucleus and binds to nucleolin and heterogeneous nuclear ribonucleoprotein A1.** Nuclear and cytosolic subcellular fractions were obtained from Jurkat cells and tested for *CD39-AS* RNA expression. **a** Mean ± SEM *CD39-AS* RNA levels in $n = 6$ Jurkat cell replicates (*$P = 0.042$ using two-sided unpaired $t$ test). In order to identify the potential binding proteins to the antisense RNA, RNA pulldown was carried out, followed by mass spectrometry. **b** Proteins identified by mass spectrometry as binders of *ENST00000452728.5* and *ENST00000414006.2* antisense splice variants are shown. Nucleolin (NCL) and heterogeneous nuclear ribonucleoprotein A1 (HNRNPA1) were the proteins showing the highest number of total spectra counts for both splice variants. **c**, **d** Mean ± SEM *CD39* mRNA and *CD39-AS* RNA levels and MFI of Jurkat cells in the absence or presence of siRNA specific to NCL, HNRNPA1, NCL plus HNRNPA1, or scramble. Exposure of Jurkat cells to NCL and HNRNPA1 siRNAs results in increased *CD39* mRNA levels and MFI (*CD39* mRNA $n = 9$ replicates; *CD39 AS* RNA $n = 8$ replicates; CD39 MFI $n = 8$ replicates) (**c** ***$P = 0.0002$, *$P = 0.04$; **d** *$P = 0.015$, **$P = 0.009$ using one-way ANOVA followed by Tukey's multiple comparisons test).

analyzed using the FlowJo 2 software (version 10, TreeStar, Ashland, OR). Fluorescence compensation was adjusted based on fluorescence-minus-one method.

**BaseScope and RNAScope**. Presence of *CD39-AS* splice variants and *CD39* mRNA transcripts was detected in Jurkat, Treg, and Th17 cells using BaseScope and RNAScope chromogenic assays (ACDBio, Newark, CA). Cells were initially fixed with 10% non-buffered formalin, washed with 1× phosphate-buffered saline, resuspended in 70% ethanol, incubated at room temperature for 10 min, and kept at 4 °C until cytospins were prepared. BaseScope and RNAScope probes were constructed by the manufacturer based on the sequence of primers used for qPCR (Supplementary Table 1). Staining was carried out according to the manufacturer's recommendations. Positively stained cells were subsequently visualized by light microscopy.

**Reverse transcription followed by qPCR**. Expression of human *CD39-AS* and *CD39* mRNA was determined by qRT-PCR. Total RNA was extracted from $3 \times 10^5$ to $5 \times 10^5$ cells using TRIzol reagent (Thermo Fisher Scientific) and mRNA was reverse transcribed using the iScript cDNA Synthesis Kit (Bio-Rad Laboratories, Hercules, CA) according to the manufacturer's instructions. CD39 and antisense primer sequences are reported in Supplementary Table 1. Samples were run on a StepOne Plus (Applied Biosystems, Foster City, CA), and results were analyzed by matched software and expressed as relative quantification. Relative gene expression was determined after normalization to human β-actin.

**Inhibition studies using FANA-CD39-AS oligonucleotides**. The effect of antisense inhibition on *CD39* mRNA levels and MFI was tested in Jurkat, Treg, and Th17 cells upon cell incubation with FANA-CD39-AS oligonucleotides (AUM Bio Tech, Philadelphia, PA). Briefly, cells were exposed to 10–20 μM FANA concentrations, and after 72 h incubation at 37 °C and 5% $CO_2$, levels of *CD39-AS* and *CD39* mRNA were tested by qPCR, whereas CD39 MFI was evaluated by flow cytometry.

**Subcellular fractionation**. Nuclear and cytoplasmic subcellular fractions from Jurkat, Treg, and Th17 cells were obtained following cell resuspension in 1× hypotonic buffer containing 20 mM Tris HCl, 10 mM NaCl, and 3 mM $MgCl_2$. Cell samples were incubated on ice for 15 min. Following addition of 10% NP40, samples were centrifuged at $850 \times g$ at 4 °C for 10 min. Supernatants (containing the cytoplasmic fraction) were then transferred into fresh tubes. TRIzol was then added to both cytoplasmic and nuclear fractions to extract RNA. Nuclear and cytoplasmic subcellular fractions were validated by qPCR upon assessment of metastasis-associated lung adenocarcinoma transcript-1 (*MALAT-1*), as marker for nuclear localization, and mitochondrially encoded cytochrome B (*MT-CYB*), as marker for cytoplasmic localization. *MALAT-1* and *MT-CYB* primer sequences are indicated in Supplementary Table 1.

**RNA pulldown assay and mass spectrometry**. Binding of *CD39-AS* RNA to nuclear proteins was detected by RNA pulldown assay and mass spectrometry. RNA pulldown was performed according to the protocol by Ruan et al.[35] with some modifications. Briefly, DNA templates were amplified from plasmids containing the two antisense variants using KOD Xtreme™ Hot Start DNA Polymerase (Millipore Sigma, Burlington, MA). PCR products were then purified from gel and biotin-labeled RNA was in vitro transcribed using the RNAMaxx High Yield Transcription Kit according to the manufacturer's protocol. RNA was purified using the RNeasy Cleanup Kit (Qiagen, Germantown, MD) and integrity and quality were verified. Nuclear lysates were prepared from Jurkat cells; proper folding of *CD39-AS* splice variant RNA baits was performed and added to Jurkat cell nuclear lysates. Following incubation with Dynabeads™ M-270 Streptavidin (Life Technologies) and washes, interacting proteins were eluted and samples were analyzed by mass spectrometry.

**CD39, NCL, and HNRNPA1 RNAi**. Jurkat, Treg, and Th17 cells were exposed to NCL and HNRNPA1 (all) and CD39 (Treg) Silencer Select siRNA (Thermo Fisher Scientific). Cells were resuspended in Opti-MEM medium (Thermo Fisher Scientific) and seeded at $1.0–1.5 \times 10^5$/well in a 96-well plate[20]. NCL, HNRNPA1, and

CD39-specific siRNA were used at a final concentration of 1 pmol/well and added to Jurkat cells or differentiated Treg and Th17 cells for the last 14 h of culture. A negative control siRNA (scramble, Thermo Fisher Scientific) served as control. NCL, HNRNPA1, and CD39 silencing was verified by qPCR using gene-specific primers following RT (Supplementary Table 1).

**Suppression assay**. Treg ability to suppress the proliferation of $CD4^+CD25^-$ cells was assessed in co-culture experiments, in which untreated, CD39 siRNA, FANA-CD39-AS oligonucleotide, NCL siRNA, HNRNPA1 siRNA, or negative control siRNA-treated Treg were added at 1:8 ratio to autologous $CD4^+CD25^-$ cells[6]. $CD4^+CD25^{high}CD127^-$ Treg and $CD4^+CD25^-$ effector cells were sorted by immunomagnetic beads (Miltenyi Biotec). Parallel cultures of $CD4^+CD25^-$ cells without Treg were performed under identical conditions. Responder cells were activated using IL2 (100 IU/ml) and Dynabeads Human T activator CD3/CD28 (bead/cell ratio: 1:2). For the last 18 h of culture, cells were pulsed with 0.25 μCi/well $^3$H-thymidine; incorporated thymidine was measured by liquid scintillation spectroscopy. The percentage of inhibition of cell proliferation was calculated using the formula [1 − count per minute (cpm) in the presence of untreated or treated Treg/cpm in the absence of Treg].

**Induction and assessment of colitis**. Colitis was induced in NOD/scid/gamma mice, pre-emptively transferred with antisense$^+$ human CD4 cells obtained from one healthy blood donor, using TNBS[28] to mostly reflect T cell-mediated tissue inflammation[36,37].

Briefly, 6-week old female NOD/scid/gamma mice were purchased from The Jackson Laboratory (Bar Harbor, ME) and kept under pathogen-free conditions between 21 °C and 23 °C, with a 12:12-h dark/light cycle and relative humidity ranging from 45 to 55%. Mice were injected with $2 \times 10^6$ human CD4 T cells obtained from one healthy blood donor. Prior to injection, cells were tested for antisense positivity by qPCR (Fig. 4b and Supplementary Fig. 7a). Three weeks after injection, mice were bled and checked for human chimerism. Mice showing >10% human chimerism were sensitized to TNBS (a kind gift from Dr. Martin Camilia R, BIDMC, Boston)[28]. One week after sensitization, mice were anesthetized and subsequently administered a single enema of 0.25 mg TNBS in 50% ethanol in a total volume of 50 μl[28]. At the time of TNBS administration, mice were given a single injection of FANA-CD39-AS oligonucleotides at 5.4 mg/kg (or vehicle) intraperitoneally, based on the manufacturer's recommendations. FANA-CD39-AS oligonucleotides specific for the two antisense variants were used and recipients were randomly allocated to treatment with FANA-CD39-AS oligonucleotides targeting either the v1 or v2 variant. After recovery from anesthesia, mice were returned to their original cage. Weight and stool were assessed and recorded daily until the day of harvest, 72 h after TNBS administration. Disease activity index was calculated on the basis of body weight loss, presence of gross blood, and stool consistency[20,28]. On the day of harvest, colons were dissected and length measured from the ileocecal junction to the anal verge. Colitis histology score was calculated after hematoxylin and eosin staining[8,9]. Peripheral blood was withdrawn and spleen, MLNs, IELs, and LPs were collected for subsequent lymphocyte analysis.

Mononuclear cells were isolated from the peripheral blood, spleen, MLN, IEL, and LP lymphocytes[8]. Mononuclear cell phenotype was then assessed by flow cytometry following exclusion of dead cells using 7-amino-actinomycin D viability staining solution (Biolegend) and gating of single cells. Lymphocyte staining was then carried out using PerCP Cy5.5 anti-human CD4 (clone # A161A1, Biolegend), FITC anti-human CD39 (clone # A1, Biolegend), and PE-Cy7 anti-human CD25 (clone # M-A251, Biolegend). Expression of FOXP3 and RORC was determined as indicated above using Pacific Blue anti-human FOXP3 (clone # 206D, Biolegend) and APC anti-human RORC (clone # AFKJS-9, eBioscience). Frequency of cytokine-producing cells was determined as indicated above. Staining was carried out using Brilliant Violet 605 anti-human IL17 (clone # BL168, Biolegend), PE anti-human IL10 (clone # JES3-19F1, Biolegend), and Alexa Fluor 700 anti-human IFNγ (clone B27, BD Biosciences). PerCP Cy5.5 rat IgG2b, κ (clone # RTK4530), FITC, Pacific Blue and Brilliant Violet 605 mouse IgG1, κ (clone # MOPC-21), and PE rat IgG2a, κ (clone # RTK2758) were used as isotype control antibodies (all from Biolegend) to define positively stained cell populations.

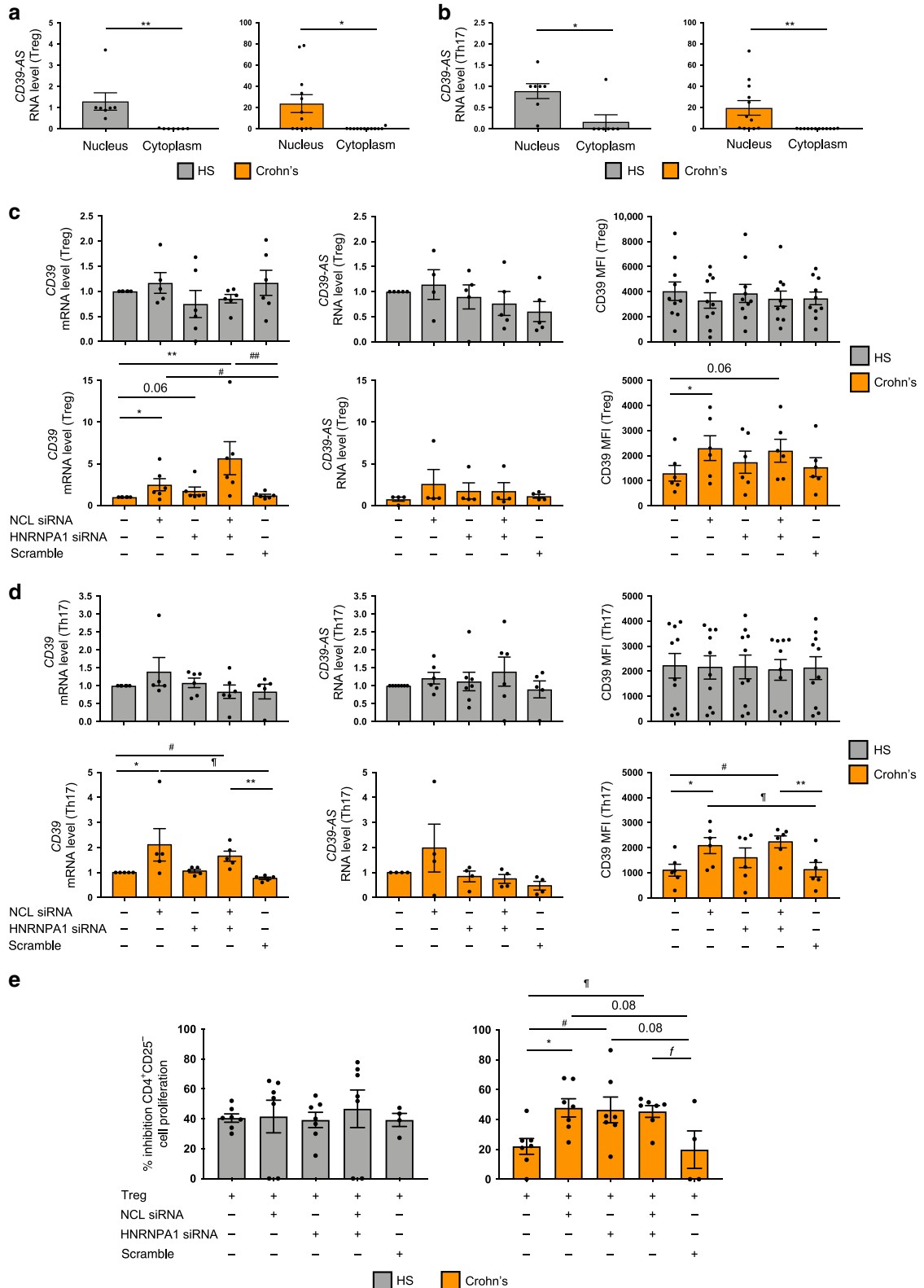

Results obtained from mice treated with FANA-CD39-AS oligonucleotides specific for either v1 or v2 variant were pooled.

An additional set of experiments was performed to determine the effect of untreated or FANA-CD39-AS-treated Treg on the course of TNBS colitis, induced in NOD/scid/gamma mice reconstituted with $2 \times 10^6$ human $CD4^+$ cells (see above). Treg were immunomagnetically sorted from the peripheral blood of the same donor as $CD4^+CD25^{high}CD127^{low}$ cells and exposed to 10 μM FANA-CD39-AS oligonucleotides prior to injection. In all, $2.5 \times 10^5$ untreated or FANA-CD39-AS-treated Treg were injected intraperitoneally at the time of TNBS

administration. Weight and stool were assessed daily and mice were harvested 72 h later. Disease activity index, colon length, and histology score were determined as indicated above.

Animal protocols were approved by the Animal Care and Use Committee at BIDMC, Boston, MA.

**Statistics**. Data were collected using Microsoft Excel for Mac (version 16.16.22). Results are expressed as mean ± SEM. Normality of variable distribution was

**Fig. 6 Silencing of NCL alone or in combination with HNRNPA1 boosts CD39 levels in Crohn's derived Treg and Th17 cells.** Nuclear and cytoplasmic subcellular fractions were obtained from Treg and Th17 cells and tested for antisense expression. **a, b** Mean ± SEM *CD39-AS* RNA levels in Treg and Th17 cells from HS ($n = 7$) and Crohn's disease patients ($n = 12$) (**a** **$P = 0.008$ for HS and *$P = 0.01$ for Crohn's patients; **b** *$P = 0.01$ for HS and **$P = 0.009$ for Crohn's patients using two-sided unpaired *t* test). Based on the results of mass spectrometry, Treg and Th17 cells, differentiated from the peripheral blood of healthy subjects and Crohn's disease patients, were exposed to scramble or gene-specific siRNA. **c, d** Mean ± SEM *CD39* and *CD39-AS* RNA levels and CD39 MFI in untreated and siRNA (or scramble) treated Treg and Th17 cells from healthy subjects (HS; Treg: *CD39* mRNA $n = 6$; *CD39-AS* RNA $n = 5$; CD39 MFI $n = 10$; Th17: *CD39* mRNA $n = 6$; *CD39-AS* RNA $n = 7$; CD39 MFI $n = 10$) and Crohn's disease patients (Treg: *CD39* mRNA $n = 6$; *CD39-AS* RNA $n = 4$; CD39 MFI $n = 6$; Th17: *CD39* mRNA $n = 5$; *CD39-AS* RNA $n = 4$; CD39 MFI $n = 6$) (**c** *$P = 0.038$, **$P = 0.002$, #$P = 0.04$, ##$P = 0.006$ for *CD39* mRNA and *$P = 0.04$ for CD39MFI; **d** *$P = 0.049$, **$P = 0.0028$, #$P = 0.012$, ¶$P = 0.01$ for *CD39* mRNA and *$P = 0.049$, **$P = 0.009$, #$P = 0.018$, ¶$P = 0.011$ for CD39 MFI using one-way ANOVA followed by Tukey's multiple comparisons test). Silencing of NCL, alone or in combination with HNRNPA1, resulted in increased *CD39* mRNA levels and MFI in Treg and Th17 cells obtained from Crohn's patients. **e** Mean ± SEM percentage inhibition of CD4[+]CD25[−] cell proliferation in the presence of untreated and siRNA (or scramble) treated Treg in HS ($n = 7$) and Crohn's disease patients ($n = 7$) (*$P = 0.029$, #$P = 0.032$, ¶$P = 0.014$, and f$P = 0.037$ using one-way ANOVA followed by Tukey's multiple comparisons test). Silencing of NCL, HNRNPA1, or both enhanced the suppressive function of Treg obtained from patients with Crohn's disease.

assessed by Kolmogorov–Smirnov goodness-of-fit test. Comparisons were performed using parametric (paired or unpaired Student's *t* test) or non-parametric (Mann–Whitney test) test according to data distribution (all two-sided). One-way analysis of variance, followed by Tukey's multiple comparisons test, was used when comparing more than two sets of data. For all comparisons, $P < 0.05$ was considered significant. $P < 0.1$ was considered to indicate a trend to significance. Statistical analysis was performed using SPSS version 22 and GraphPad Prism, version 7.0e.

**Reporting summary**. Further information on research design is available in the Nature Research Reporting Summary linked to this article.

## Data availability
Source data are provided with this paper. For further request, please contact the corresponding author.

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

## Acknowledgements

This work has been supported by the National Institute of Health (R01 DK108894 to M.S.L. and R21 CA164970 to S.C.R.), AASLD Pilot Research Award (to M.S.L.), Seed Grant Award (Department of Anesthesia, Critical Care & Pain Medicine to M.S.L.), the Department of Defense Award W81XWH-16-0464 (to S.C.R.), Postdoctoral Fellowship from CAPES, Brazil (88881.171857/2018-01 to L.A.F.), Scholarship Dr. Ludmily Sedlarovey-Rabanovej and Nadacia Tatra banky and the Slovak Research and Development Agency under the contract no. APVV-17-0505 (to B.G.). The authors wish to thank Dr. Alan C. Mullen (Harvard Stem Cell Institute, Massachusetts General Hospital, Boston, MA) and Dr. David J. Friedman (Beth Israel Deaconess Medical Center, Boston, MA) for helpful discussions and critical revision of the manuscript.

## Author contributions

R.P.H., A.X., M.V.: acquisition of data, analysis and interpretation of data, drafting of the manuscript; L.A.F., B.G., H.Z., R.J.R., E.C.: acquisition of data; S.M.: analysis and interpretation of data; E.K., A.S.C., A.C.M., S.K.K., S.C.R., M.S.L.: critical revision of the manuscript; M.S.L.: writing of the manuscript; M.S.L., S.C.R.: obtained funding.

## Competing interests

The authors declare no competing interests.
