## [Peer Review File · Nature Communications]

Reviewers' comments:

Reviewer #1 (Remarks to the Author):

The authors demonstrated that CD39-specific antisense is increased in Treg and Th17 in Crohn's disease patients, silence this antisense restore the expression of CD39 and ameliorate disease activity in experimental colitis which lead to a potential novel therapeutic strategy of restoring CD39 expression in Crohn's disease by inhibiting CD39-specific antisense. The design and aims of this study are clear. The results are insightful.

However, some issues still need to be addressed comprehensively, the main issue is that the authors indicated that CD39-specific antisense regulated CD39 expression which resulted in immune balance between Treg and Th17 in the disease of IBD, however how does Treg and Th17 been modified by CD39-specific antisense was not been clearly investigated:

1. What is the relationship between CD39 expression and disease severity in the clinic and the correlation of CD39 expression to Tregs and Th17 in PBMCs.
2. The authors demonstrated the differences in CD39-AS and CD39 expression in Treg and Th17 cells between healthy people and Crohn's disease patients (Fig.1, Fig.2 and Fig.3). However, there was no significant differences of CD4+Foxp3+ cells and CD4+IL-10+ cells between vehicle and FANA group in experimental colitis models. We all know that Treg cells play a pivotal immune-suppressive role in autoimmune disease like Crohn's disease one of their weapons is IL-10. Please explain this result.
3. What is the effect of CD39-AS and CD39 expression to the immune regulatory function or protein (Foxp3, IL-17) expression of Treg and Th17?
4. The authors clearly showed that FANA can restore CD39 expression in Treg cells. However, we still do not know if FANA treatment affect Treg cells' suppressive function. Perhaps a CFSE test will make the entire study more complete.
5. Please show dot plot for figure 3 and Foxp3 expression on different time point with FANA treatment in figure 3.
6. Please show gating strategy and dot plot for figure 4G, and level of CD39 expression in CD4+Foxp3+ and CD4+IL-17+ cells.
7. The authors clearly showed that siRNA for Nucl and HNRNPA1 can restore CD39 expression in Treg cells. So what is the effect of the siRNA to the function of Tregs.
8. Other Th cells like Th1 cells also play an important role in Crohn's disease. So does CD39-AS have a regulatory effect on Th1 cells?

Reviewer #2 (Remarks to the Author):

The manuscript entitled "Endogenous antisense RNA curbs CD39 expression in Crohn's disease" by Harshe et al. summarize the expression and role of non-coding antisense RNA in Th17 T cells and Tregs and investigate their contribution to Crohn's disease and suggest the possibility of the therapeutic use of these antisense oligonucleotides. Because treatment options for Crohn's disease are limited, new strategies for the treatment of Crohn's disease is of significance.

However, it is well known that apart from T cells, various cells within the immune system such as B-cells, dendritic cells, NK cells, monocytes, macrophages as well as endothelial cells in the colon express CD39. The majority of these cells play a role during mucosal inflammations/colitis. Why the authors only focus on Treg and Th17 remains unclear.

The possibility of ATP produced by gut bacteria that may a critical role especially in Th17 T-cell immune deviation in the lamina propria is not mentioned nor discussed. (Atarashi K, et al. (2008) ATP drives lamina propria T(H)17 cell differentiation. Nature 455:808-812). Distinguishing

between host- and microbial-derived ATP in the context of IBD could further emphasize the importance of CD39 in IBD.

Major points

1. Although it is nicely described in the introduction section, the first line of the abstract is an incomplete statement due to the fact that CD73 is required for the conversion of AMP to adenosine.
2. Considering the complexity of extracellular nucleotide signaling that includes P1 receptors, P2X ligand-gated ion channels, P2Y GPCRs, receptors with enzymatic activity that metabolize nucleotides and variety of hematopoietic and non-hematopoietic cells that bear a different collection of these molecules, authors should better rationalize the reasons for only focussing on T-regs and Th17 cells. At least, Th1 and Th2 cells need to be investigated.
3. Isolating and polarizing human T cells are a very common practice, using them for all functional experiments will give a better understanding. Instead of using Jurkat cells, primary human T cells are of more physiological relevance.
4. There is a difference in the size of sequences in the manuscript and existing database information.
ENST00000414006 (henceforth v2) is 2457 bp
http://www.ensembl.org/Homo_sapiens/Transcript/Exons?db=core;g=ENSG00000226688;r=10:95732976-96090250;t=ENST00000414006
In the manuscript, it is stated as 591 bp.
5. It is stated: "To validate the expression dynamics of CD39-AS RNA in T cells, reverse transcription followed by qPCR (RT-qPCR) was performed on RNA isolated from Jurkat and peripheral blood-derived human T cells...". But in respective figures, there is only data from cell lines. Please use also human T cells.
6. Using Jurkat cells as a positive control for CD39-AS is fine but authors should comment/discuss the differences between the four cell lines. It is obvious that there are additional mechanisms other than CD39-AS in regulating CD39 expression in each cell type.
7. An additional table showing relevant clinical information (treatments, scores, etc.) of healthy controls and patients would be helpful.
8. Figure 2E, only CD39-AS profile of immunosuppressant treated subjects were shown, we don't know if this significant reduction leads to an increase of CD39 upon treatment.
9. Figure 2F. This graph is misleading. Figure legends state that CD39-AS was correlated with disease activity (Harvey-Bradshaw-Index). However, the x-axis states Montreal (disease location).
10. Figure 2G, In CD39 results for the non-inflamed group, there is one value which is 20x bigger than the rest of the group, which is outside of normal distribution. Analyzing more samples for this group is necessary.
11. Figure 3 A, B, C, As mentioned before using Jurkat cells as positive is fine, but the authors should have used polarized human T cells to determine the optimal time point for FANA oligonucleotide application.
12. The control oligonucleotide used as a control during in vitro experiments would have been better control for this experiment instead of a vehicle for example in Figure 3A.
13. Figure 4B. It is not clear why B-actin bands are separated and CD39 bands are not.

14. Figure 4F. It is not clear which CD4+ cells are shown. Please show the entire gating strategies. Also, use better dot plots instead of density or contour plots.

15. Figure 4G. A statement about decreased IL17 producing CD4 cells in MLN is not consistent with the figure.

16. Since there is no difference in frequency of CD39+, IL17+, IL10+, and FoxP3+ CD4 T cells, isolated from LP, MLN, and IEL, how the authors explain that the beneficial effect of FANA CD39-AS in TNBS model is only through Th17 or T-regs cells?

17. The authors need to discuss, why differences are only observed in spleen and blood but not in the lamina propria, where the inflammation is located.

18. An additional colitis model would be useful. (Transfer colitis with Naïve T cells +/- CD39 KO Tregs? + CD39-AS administration).

19. The rationale for choosing the TNBS colitis model is not given.

15. In Figure 5, the experimental setup to show the localisation of CD39-AS and interacting proteins is well conducted. The identification of nuclear proteins using human Th17 or Treg cells would be more relevant to better translate the results to possible therapeutic options for patients with Crohn's Disease (other diseases as well).

16. The two candidates NCL and HNRNPA1 were identified upon binding to CD39-AS but apparently their absence failed to modulate CD39-AS expression. It seems to block those genes that resulted in increased CD39 expression in a CD39-AS independent manner.

17. Supplementary Tables 2 and 3 were not included in the file

Minor:

1. In figure 1 all of the three antisense products are named AS1 and this makes it difficult to follow the manuscript.

2. CD39-AS1 primers amplify two different products 135 bp and 257 bp, the amplicon size difference could be better emphasized in figure 1A.

3. Supplementary data: Antisense variant 2 sequence includes lowercase bases.

4. Supplementary Fig. 1: Legend: "cells from one healthy subject and one AIH patient are shown". In the figure, only one of them is shown.

Point by point reply to Reviewers

Reviewer # 1

The authors demonstrated that CD39-specific antisense is increased in Treg and Th17 in Crohn's disease patients, silence this antisense restore the expression of CD39 and ameliorate disease activity in experimental colitis which lead to a potential novel therapeutic strategy of restoring CD39 expression in Crohn's disease by inhibiting CD39-specific antisense. The design and aims of this study are clear. The results are insightful.

However, some issues still need to be addressed comprehensively, the main issue is that the authors indicated that CD39-specific antisense regulated CD39 expression which resulted in immune balance between Treg and Th17 in the disease of IBD, however how does Treg and Th17 been modified by CD39-specific antisense was not been clearly investigated:

Author's reply: We thank the Reviewer for the comment and observation. In this revised version of the manuscript we provide evidence that CD39 antisense (CD39-AS) RNA impacts the functionality of regulatory T cells (Treg). Our data indicate that antisense silencing achieved upon cell exposure to FANA-CD39-AS oligonucleotides boosts Treg ability to suppress CD4⁺CD25⁻ responder cell proliferation in vitro (Fig. 3F); whereas Treg exposure to CD39 silencing dampens Treg ability to suppress effector cell proliferation (Figure 3F). Further, analysis of the effects of FANA-CD39-AS oligonucleotide administration in vivo, in a humanized mouse model of TNBS-induced colitis, has shown increase in the expression of CD39 levels in peripheral blood CD4⁺FOXP3⁺, CD4⁺IL17⁺ and CD4⁺IL10⁺ cells (Supplementary Fig. 8F). Notably, injection of FANA-treated human Treg into NOD/scid/gamma mice, previously reconstituted with human CD4 cells and administered TNBS, resulted in amelioration of colitis (Supplementary Fig. 9).

1. What is the relationship between CD39 expression and disease severity in the clinic and the correlation of CD39 expression to Tregs and Th17 in PBMCs.

Author's reply: We have previously shown that single nucleotide polymorphisms in noncoding regions of the ENTPD1 gene are associated with low levels of CD39 mRNA and with predisposition to Crohn's disease in humans (Friedman et al, 2009 PMID:19805374). Importance of ENTPD1/CD39 in the context of colitis has been also corroborated by studies in experimental colitis, in which we have shown that Cd39^{-/-} mice exposed to dextran-sulfate-sodium (DSS) develop a more severe form of colitis than wild type counterparts (Friedman et al, 2009 PMID:19805374). In additional studies by us and others, CD39 was found pivotal to Treg function through the generation of adenosine (Deaglio et al, 2007 PMID: 17502665; Borsellino et al, 2007 PMID: 17449799; Grant et al, 2014 PMID: 23787765). Further, heightened expression of CD39 by Treg was previously associated with therapeutic remission in IBD patients (Gibson et al, 2015 PMID: 26332314); while overexpression of this ectoenzyme resulted in Treg more effective at controlling disease activity in a T-cell transfer model of colitis (Robles et al, 2019 PMID: 31693091). When considering Th17 cells, we have found that expression of CD39 confers immunoregulatory properties to these cells and that these CD39⁺ 'suppressor' Th17 cells are impaired in patients with Crohn's disease, as result of aberrant aryl hydrocarbon receptor and hypoxia-inducible-

factor-1alpha signaling pathways (Longhi et al. 2014 PMID: 24505337; Longhi et al, 2017 PMID: 28469075; Xie et al, 2018 PMID: 30098863).

All these references have been quoted in the revised version of our manuscript (see Introduction). As a further support to the correlation between CD39 levels and disease activity, Fig. 2G in the current version of the manuscript shows significantly higher levels of CD39 in Th17 cells obtained from non-inflamed as compared to inflamed biopsied areas of Crohn's disease patients. Higher CD39 levels are also noted in Treg derived from non-inflamed biopsied areas and this increase trended to significance (Fig. 2G).

2. The authors demonstrated the differences in CD39-AS and CD39 expression in Treg and Th17 cells between healthy people and Crohn's disease patients (Fig. 1, Fig. 2 and Fig. 3). However, there was no significant differences of CD4⁺Foxp3⁺ cells and CD4⁺IL-10⁺ cells between vehicle and FANA group in experimental colitis models. We all know that Treg cells play a pivotal immune-suppressive role in autoimmune disease like Crohn's disease one of their weapons is IL-10. Please explain this result.

Author's reply: We thank the Reviewer for the observation. Our data indicate that there is no effect of FANA treatment on the frequency of CD4⁺FOXP3⁺ cells in the experimental colitis model (Figure 4G). These findings are supported also by in vitro data, showing that treatment with FANA oligonucleotides does not change FOXP3 MFI in Treg polarized from peripheral blood CD4 lymphocytes (Supplementary Fig. 6B). Importantly, however, we do find that FANA treatment: a) boosts CD39 MFI in peripheral blood CD4⁺FOXP3⁺, CD4⁺IL10⁺ and CD4⁺IL17⁺ cells in the experimental colitis model in vivo (Supplementary Fig. 8F); b) renders Treg, isolated from the peripheral blood of Crohn's disease patients, more effective at suppressing CD4⁺CD25⁻ cell proliferation (Fig. 3F); please note that in the same experiment silencing of CD39 results in marked decrease in Treg suppressive abilities in both patients and healthy controls (Fig. 3F); c) boosts Treg ability to control disease activity in vivo in a model of experimental colitis in NOD/scid/gamma humanized mice. As shown in Supplementary Fig. 9, NOD/scid/gamma recipients initially reconstituted with human CD4 cells and subsequently injected with FANA-treated autologous Treg at the time of TNBS administration displayed the lowest disease activity index, the highest colon length and the lowest histology score.

3. What is the effect of CD39-AS and CD39 expression to the immune regulatory function or protein (Foxp3, IL-17) expression of Treg and Th17?

Author's reply: As indicated above, treatment with FANA-CD39-AS oligonucleotides boosts the in vitro suppressive function of Treg derived from Crohn's disease patients (Fig. 3F). The fact that exposure to FANA oligonucleotide treatment does not result in significant amelioration of the suppressive function of Treg isolated from healthy subjects might relate to the fact that CD39-AS RNA levels in health are lower, when compared to Crohn's disease patients; therefore, silencing the antisense in these cells might not result in marked effects on their function. Our new findings also indicate that injection of FANA-CD39-AS oligonucleotide treated Treg into humanized NOD/scid/gamma mice results in the lowest disease activity index, greatest colon length and markedly reduced histology score (Supplementary Fig. 9). Despite not having effect on Treg FOXP3 levels in vitro and on the frequency of CD4⁺FOXP3⁺ cells in vivo, we note that FANA treatment boosts CD39 MFI in peripheral blood CD4⁺FOXP3⁺, CD4⁺IL17⁺ cells and CD4⁺IL10⁺

cells *in vivo* (Supplementary Fig. 8F); further it decreases the proportion of CD4⁺IL17⁺ cells in the spleen and peripheral blood and, although to a lesser extent, IEL (Fig. 4G).

4. The authors clearly showed that FANA can restore CD39 expression in Treg cells. However, we still do not know if FANA treatment affect Treg cells' suppressive function. Perhaps a CFSE test will make the entire study more complete.

Author's reply: Please see answer to point 4. As we previously found that ³H-thymidine incorporation-based assay and CFSE give comparable results in suppression assay experiments, the ³H-thymidine incorporation assay was chosen to assess CD4⁺CD25⁻ cell proliferation, in the absence or presence of Treg, given the requirement for fewer cells (Grant et al, 2009 PMID: 23787765).

5. Please show dot plot for figure 3 and Foxp3 expression on different time point with FANA treatment in figure 3.

Author's reply: Representative dot plots of forward scatter and CD39 fluorescence in Jurkat and healthy control-derived Treg and Th17 cells before and after FANA-CD39-AS oligonucleotide treatment at 24, 48 and 72 hours are provided in Supplementary Fig. 3A-B. Cumulative data of CD39 MFI in healthy control derived Treg and Th17 cells at different time points are provided in Supplementary Fig. 4C.

Due to low numbers of Th17 and Treg derived from the peripheral blood of Crohn's disease patients, analysis of FANA treatment was done only at baseline and at 72 hours. Representative dot plots of Treg and Th17 cells before and after FANA-CD39-AS treatment are presented in Supplementary Fig. 3C.

Cumulative data of FOXP3 MFI in Jurkat and healthy control-derived Treg and Th17 cells before and after exposure to FANA-CD39-AS oligonucleotide treatment at 24, 48 and 72 hours are shown in Supplementary Fig. 6A-B. Due to the relatively low number of Treg and Th17 cells obtained from patients' CD4 cells, FOXP3 staining could not be performed in cell subsets from this group.

6. Please show gating strategy and dot plot for figure 4G, and level of CD39 expression in CD4⁺Foxp3⁺ and CD4⁺IL-17⁺ cells.

Author's reply: The gating strategy and representative dot plots for Figure 4G are provided in Supplementary Fig. 8A-E. CD39 MFI for CD4⁺FOXP3⁺ and CD4⁺IL17⁺ cells are shown in Supplementary Fig. 8F.

7. The authors clearly showed that siRNA for Nucl and HNRNPA1 can restore CD39 expression in Treg cells. So what is the effect of the siRNA to the function of Tregs.

Author's reply: We thank the Reviewer for the comment. In addition to restoring CD39 levels in Treg obtained from Crohn's disease patient samples, silencing of NCL and HNRNPA1 boosts Treg suppressive function in these cells. These new findings are shown in Fig. 6E. As for qPCR and FACS data, these results are noted for Crohn's but not healthy control derived Treg cells; this suggesting that there might be intrinsic differences in healthy versus Crohn's disease Treg

lymphocytes in the formation of CD39-AS RNA-protein complexes, as commented in the Discussion (line 342-352).

8. Other Th cells like Th1 cells also play an important role in Crohn's disease. So does CD39-AS have a regulatory effect on Th1 cells?

Author's reply: We thank the Reviewer for the comment. We have obtained Th1 and Th2 cells (the latter in response to a similar point raised by Reviewer 2) upon polarization of peripheral blood CD4 cells of healthy controls and Crohn's disease patients. Both Th1 and Th2 cells express markedly low levels of CD39-AS RNA (see Supplementary Fig. 2) and treatment of these cells with FANA-CD39-AS oligonucleotides does not significantly alter CD39-AS RNA, CD39 mRNA (Supplementary Fig. 5A-B), CD39 MFI (Supplementary Fig. 5C-D) along with IFN γ (in Th1) and IL4 (in Th2) levels (Supplementary Fig. 5C-D), in both controls and patients.

Reviewer # 2

The manuscript entitled "Endogenous antisense RNA curbs CD39 expression in Crohn's disease" by Harshe et al. summarize the expression and role of non-coding antisense RNA in Th17 T cells and Tregs and investigate their contribution to Crohn's disease and suggest the possibility of the therapeutic use of these antisense oligonucleotides. Because treatment options for Crohn's disease are limited, new strategies for the treatment of Crohn's disease is of significance.

However, it is well known that apart from T cells, various cells within the immune system such as B-cells, dendritic cells, NK cells, monocytes, macrophages as well as endothelial cells in the colon express CD39. The majority of these cells play a role during mucosal inflammations/colitis. Why the authors only focus on Treg and Th17 remains unclear.

Author's reply: We thank the Reviewer for giving us the opportunity to comment on this. We are aware of the evidence indicating expression of CD39 in multiple immune and non-immune cell subsets, all contributing - at least to some extent - to IBD pathogenesis. The choice of focusing our studies on Treg and Th17 cells was based upon: a) initial experiments aimed at measuring levels of CD39 mRNA and CD39-AS RNA in cell lines of different lineages and indicating that Jurkat cells were the cell line with the highest CD39-AS RNA expression along with low CD39 mRNA levels (Fig. 1B-C), this suggesting that the antisense based mechanism of CD39 regulation might have the largest impact on T lymphocytes; b) our previously published evidence indicating marked CD39 impairment in Treg and Th17 cells obtained from Crohn's disease patients along with alterations in pathways contributing to CD39 regulation in the setting of Th17 cells, i.e. aryl hydrocarbon receptor and hypoxia inducible factor 1-alpha signaling pathways (Gibson et al, 2015 PMID: 26332314; Longhi et al, 2014 PMID: 24505337; Longhi et al, 2017 PMID: 28469075; Xie et al, 2018 PMID: 30098863). In this revised manuscript, we have also extended our analysis of CD39 and CD39-AS RNA to other T cell subsets, namely Th1 and Th2 lymphocytes, known to play a role in Crohn's disease immunopathogenesis. These new studies, however, show that the levels of CD39-AS RNA and CD39 mRNA are markedly low in both Th1 and Th2 subsets (Supplementary Fig. 2A-B), postulating that regulation of CD39 by the antisense RNA is predominant in Treg and Th17 cell subsets.

The possibility of ATP produced by gut bacteria that may a critical role especially in Th17 T-cell

immune deviation in the lamina propria is not mentioned nor discussed. (Atarashi K, et al. (2008) ATP drives lamina propria T(H)17 cell differentiation. Nature 455:808–812). Distinguishing between host- and microbial-derived ATP in the context of IBD could further emphasize the importance of CD39 in IBD.

Author's reply: We are grateful to the Reviewer for this important comment and observation. We have now quoted and commented the paper by Atarashi et al. in the Discussion (line 276-281).

Major points

1. Although it is nicely described in the introduction section, the first line of the abstract is an incomplete statement due to the fact that CD73 is required for the conversion of AMP to adenosine.

Author's reply: We thank the Reviewer for pointing this out. We have now amended the first sentence of the abstract, as suggested.

2. Considering the complexity of extracellular nucleotide signaling that includes P1 receptors, P2X ligand-gated ion channels, P2Y GPCRs, receptors with enzymatic activity that metabolize nucleotides and variety of hematopoietic and non-hematopoietic cells that bear a different collection of these molecules, authors should better rationalize the reasons for only focussing on T-regs and Th17 cells. At least, Th1 and Th2 cells need to be investigated.

Author's reply: We thank the Reviewer for the comment. With regard to the reasons for focusing on Treg and Th17 cells, please see answer to comment 1. We have now included novel investigations on Th1 and Th2 cells, as recommended (please see results presented in Supplementary Fig. 1C-D; Supplementary Fig. 2A-B; and Supplementary Fig. 5).

3. Isolating and polarizing human T cells are a very common practice, using them for all functional experiments will give a better understanding. Instead of using Jurkat cells, primary human T cells are of more physiological relevance.

Author's reply: Thank you for your comment. Results obtained from Treg and Th17 cells polarized from peripheral blood and lamina propria derived CD4 cells are presented in Fig. 2, Fig. 3, Fig. 6, Supplementary Fig. 1, Supplementary Fig. 3, Supplementary Fig. 4, Supplementary Fig. 6 and Supplementary Fig. 7.

4. There is a difference in the size of sequences in the manuscript and existing database information.

ENST00000414006 (henceforth v2) is 2457 bp
http://www.ensembl.org/Homo_sapiens/Transcript/Exons?db=core;g=ENSG00000226688;r=10:95732976-96090250;t=ENST00000414006

In the manuscript, it is stated as 591 bp.

Author's reply: We thank the Reviewer for pointing this out. We would like to clarify that the ENST00000414006 variant we were referring to in our original submission of the manuscript is the ENST00000414006.1(591bp) annotation according to GRCh37. We agree with the reviewer

that the latest annotation of *ENST00000414006.2* variant is indeed 2.457 Kb as indicated by the Reviewer. We have now included the entire cDNA sequences corresponding to variant 1 (*ENST00000452728.5*, 858bp) and variant 2 (*ENST00000414006.2*, 2,457bp) that are analyzed in this study (see Supplemental Material).

5. It is stated:” To validate the expression dynamics of CD39-AS RNA in T cells, reverse transcription followed by qPCR (RT-qPCR) was performed on RNA isolated from Jurkat and peripheral blood-derived human T cells...”. But in respective figures, there is only data from cell lines. Please use also human T cells.

Author’s reply: Results on CD39-AS RNA expression in different subsets of human T cells are shown in Supplementary Fig. 2.

6. Using Jurkat cells as a positive control for CD39-AS is fine but authors should comment/discuss the differences between the four cell lines. It is obvious that there are additional mechanisms other than CD39-AS in regulating CD39 expression in each cell type.

Author’s reply: We thank the Reviewer for the insightful comment. The data shown in Fig. 1B-C indicate high CD39-AS RNA and low CD39 mRNA levels in Jurkat cells, suggesting that, in this cell line, CD39 expression can be regulated by CD39-AS RNA. Such correlation appears to be present, at least in part, also for the B cell line HCC1739 BL, which displays markedly high CD39 mRNA levels - a typical feature of B cells - and low CD39-AS RNA; but not in the case of Raji and THP-1 cell lines, implicating mechanisms of CD39 regulation different from the antisense in these two cell lines. This comment has now been added in the Discussion (line 292-295).

7. An additional table showing relevant clinical information (treatments, scores, etc.) of healthy controls and patients would be helpful.

Author’s reply: Clinical information of Crohn’s disease patients has been summarized in Supplementary Table 4. Since peripheral blood samples to be used as control were obtained from healthy blood donors, no clinical information such as treatments or scores is available for this group.

8. Figure 2E, only CD39-AS profile of immunosuppressant treated subjects were shown, we don’t know if this significant reduction leads to an increase of CD39 upon treatment.

Author’s reply: Data on Treg CD39 mRNA expression in the absence or presence of immunosuppressive drugs have now been added in Fig. 2E. Corresponding text and figure legend have been also updated.

9. Figure 2F. This graph is misleading. Figure legends state that CD39-AS was correlated with disease activity (Harvey-Bradshaw-Index). However, the x-axis states Montreal (disease location).

Author’s reply: Please note that legend to Figure 2F states: ‘Correlation between CD39-AS RNA levels with Montreal type in Treg, and with Harvey-Bradshaw-Index in Th17 cells’. In Panel 2F

correlations between CD39-AS RNA levels in Treg and Montreal type and between CD39-AS RNA levels in Th17 cells with Harvey Bradshaw Index are shown.

10. Figure 2G, In CD39 results for the non-inflamed group, there is one value which is 20x bigger than the rest of the group, which is outside of normal distribution. Analyzing more samples for this group is necessary.

Author's reply: Thank you for your comment. We have now included data obtained from the analysis of Treg and Th17 cells derived from additional non-inflamed biopsied areas. Graphs in Fig. 2G have been updated accordingly.

11. Figure 3 A, B, C, As mentioned before using Jurkat cells as positive is fine, but the authors should have used polarized human T cells to determine the optimal time point for FANA oligonucleotide application.

Author's reply: We thank the Reviewer for the suggestion. We have now added data on the effects of FANA-CD39-AS oligonucleotide treatment on CD39-AS RNA, CD39 mRNA, CD39 and FOXP3 MFI in human Treg and Th17 cells at different time points. These data are presented in Supplementary Fig. 4 and Supplementary Fig. 6.

12. The control oligonucleotide used as a control during in vitro experiments would have been better control for this experiment instead of a vehicle for example in Figure 3A.

Author's reply: We have now clarified that the control we have used in the experiments shown in Fig. 3 as well as Fig. 5 and Fig. 6 is 'scramble'.

13. Figure 4B. It is not clear why B-actin bands are separated and CD39 bands are not.

Author's reply: We have now re-run the gel using the same samples. In the new Fig. 4B, bands referring to CD39, CD39-AS and β -actin are all contiguous.

14. Figure 4F. It is not clear which CD4+ cells are shown. Please show the entire gating strategies. Also, use better dot plots instead of density or contour plots.

Author's reply: The cells shown in Fig. 4F are peripheral blood mononuclear cells. The density plots presented originally in this panel have now been replaced with dot plots. Gating strategy and dot plots of CD4⁺CD39⁺, CD4⁺FOXP3⁺, CD4⁺IL17⁺ and CD4⁺IL10⁺ cells in different compartments is now provided in Supplementary Fig. 8.

15. Figure 4G. A statement about decreased IL17 producing CD4 cells in MLN is not consistent with the figure.

Author's reply: We apologize for the mistake. This has now been corrected in the Results section (line 211-212) and in the Discussion (line 314-315).

16. Since there is no difference in frequency of CD39+, IL17+, IL10+, and FoxP3+ CD4 T cells, isolated from LP, MLN, and IEL, how the authors explain that the beneficial effect of FANA CD39-AS in TNBS model is only through Th17 or T-reg cells?

Author's reply: We thank the Reviewer for the observation and comment. We have conducted further analysis and found that treatment with FANA-CD39-AS oligonucleotides increases CD39 MFI in the peripheral blood of CD4⁺FOXP3⁺, CD4⁺IL17⁺ and CD4⁺IL10⁺ cells (see Supplementary Fig. 8F). Based on these data, we postulate that FANA treatment has beneficial effects in experimental colitis by boosting CD39 levels in both regulatory and effector CD4 cell compartments, rather than having a direct effect on the overall frequency of these cell subsets. This has now been emphasized in the Discussion (line 312-323).

17. The authors need to discuss, why differences are only observed in spleen and blood but not in the lamina propria, where the inflammation is located.

Author's reply: We thank the Reviewer for the comment. For completeness, we have now included a P value indicating a trend to significance when considering the frequency of CD4⁺IL17⁺ cells in the intraepithelial lymphocyte compartment in mice treated with vehicle or receiving FANA (see Fig. 4G). The overall limited effect of FANA treatment on the phenotype of CD4 lymphocytes derived from intraepithelial or lamina propria compartments suggests that the regulation of cell phenotypes in the gut might result from a combination of factors that also include cytokine milieu, interactions with other cell types and presence of additional regulatory circuits that might operate at the local level. Whether the increase in CD39 MFI within CD4⁺FOXP3⁺, CD4⁺IL17⁺ and CD4⁺IL10⁺ peripheral compartment also results in the regulation of inflammatory cell trafficking, as previously reported (Hyman et al, 2009 PMID: 19381014), will need further investigations. This comment has now been added in the discussion (line 319-323).

18. An additional colitis model would be useful. (Transfer colitis with Naïve T cells +/- CD39 KO Tregs? + CD39-AS administration).

Author's reply: As the CD39-AS RNA is only present in humans, we had to deploy humanized mice to induce colitis following reconstitution of recipients with human CD4 cells. In a new set of experiments, we have tested the effects of Tregs exposed to FANA-CD39-AS oligonucleotides on TNBS colitis induced in NOD/scid/gamma mice reconstituted with human CD4 cells. Untreated and FANA-CD39-AS oligonucleotide treated Treg were injected at the time of TNBS administration. Mice were sacrificed 72 hours later and organs harvested. We have found decrease in the disease activity index, increase in colon length and decreased histology score in recipients of FANA-CD39-AS treated Treg, as compared with mice receiving vehicle or untreated Treg. These new results are presented in Supplementary Fig. 9. We had already tested in previous studies the effect of Cd39^{-/-} Treg in adoptive transfer colitis and found that these cells were less effective at protecting animals from disease (see Gibson et al 2015 PMID: 26332314). For this reason, injection of Cd39^{-/-} cells was not pursued in the current experimental setting.

19. The rationale for choosing the TNBS colitis model is not given.

Author's reply: TNBS was chosen to mostly reflect T cell mediated colonic tissue inflammation (Wirtz et al, 2017 PMID: 28569761; Wunschel et al, 2017 PMID: 28955241); this information has now been added in the text (line 501-502).

15. In Figure 5, the experimental setup to show the localisation of CD39-AS and interacting proteins is well conducted. The identification of nuclear proteins using human Th17 or Treg cells would be more relevant to better translate the results to possible therapeutic options for patients with Crohn's Disease (other diseases as well).

Author's reply: We thank the Reviewer for the comment. We have now performed RNA pulldown assay using Treg and Th17 cells polarized from one healthy blood donor. The Treg and Th17 cell yield normally obtained from Crohn's disease patient samples does not enable achieving the amount of protein necessary to run the RNA pulldown (70-100 mg of protein per cell type). The RNA pulldown assay showed that the two antisense variants bind to the HNRNPA1 protein. Despite the total spectra count was lower than in the experiment with Jurkat cells (ranging from 1 to 4), no counts were noted for HNRNPA1 protein in the control (i.e. beads) samples, indicating specificity of the results obtained. The reasons for not detecting the NCL protein might be linked to the fact that the overall antisense expression in Treg and Th17 cells is lower in healthy controls than in Crohn's disease patients (Fig. 2A-D), therefore the amount and number of proteins bound to it could be substantially reduced in controls than in patients' samples. Despite we could not perform RNA pulldown assay on Crohn's disease derived cells due to the limitations mentioned above, the evidence that silencing of NCL and/or HNRNPA results in increased CD39 levels and Treg function (Fig. 6C-E) suggests that CD39-AS RNA interactions with both these proteins play a role in the regulation of CD39 levels in this setting. Both NCL and HNRNPA1 were expressed in Treg and Th17 nuclear cell fractions of controls and patients, despite to a higher level in the latter (see Figure below).

Mean + SEM NCL and HNRNPA1 mRNA expression in the nuclear fraction of Treg and Th17 cells derived from healthy subjects (HS) (n=4) and Crohn's disease patients (n=4). P*: P≤0.05; **: P≤0.01

16. The two candidates NCL and HNRNPA1 were identified upon binding to CD39-AS but apparently their absence failed to modulate CD39-AS expression. It seems to block those genes that resulted in increased CD39 expression in a CD39-AS independent manner.

Author's reply: We thank the Reviewer for the observation. As noted, exposure to NCL and HNRNPA1 silencing results in increased Treg and Th17 CD39 expression and - as our new data indicate - boosted Treg suppressive function in samples derived from patients with Crohn's disease. The evidence that no effect is observed on CD39-AS RNA levels after NCL and HNRNPA1 silencing does not preclude that the antisense cannot regulate CD39. As the regulation operated by the antisense partly relies on the binding with NCL and HNRNPA1, silencing of these two

proteins can interfere with the ability of CD39-AS RNA ability to curb CD39 and not with a decrease in its expression levels.

17. Supplementary Tables 2 and 3 were not included in the file

Author's reply: Supplementary Tables 2 and 3 have now been provided as word documents.

Minor:

1. In figure 1 all of the three antisense products are named AS1 and this makes it difficult to follow the manuscript.

Author's reply: In the revised Fig. 1A, we have now indicated the names and transcript ids for the splice variants according to human genome assembly GRCh38. Transcript id: ENST00000452728.5 corresponds to variant 1 (v1) and transcript id: ENST00000414006.2 corresponds to variant 2 (v2) in the manuscript text. The longer ENSG00000226688.7 variant was not analyzed further in this study. We have also now included the complete cDNA sequences and primer binding of v1 and v2 in the Supplemental Material.

2. CD39-AS1 primers amplify two different products 135 bp and 257 bp, the amplicon size difference could be better emphasized in figure 1A.

Authors reply: We thank the Reviewer for the suggestion. In the revised version, we have shown the entire ENTPD1 domain in Fig 1A. Complete cDNA sequences corresponding to ENST00000452728.5 and ENST00000414006.2 along with primer binding sites and amplified sequences were highlighted and have now been provided in the Supplemental Material.

3. Supplementary data: Antisense variant 2 sequence includes lowercase bases.

Author's reply: We have now shown the FANA antisense target sequences in the cDNA sequences corresponding to ENTPD1-AS1 splice variants, ENST00000452728.5 and ENST00000414006.2.

4. Supplementary Fig. 1: Legend: "cells from one healthy subject and one AIH patient are shown". In the figure, only one of them is shown.

Author's reply: We apologize for the typo. The legend to Supplementary Figure 1 has now been corrected.

REVIEWER COMMENTS

Reviewer #2 (Remarks to the Author):

The revised manuscript by Harshe and colleagues entitled "Endogenous antisense RNA curbs CD39 expression in Crohn's disease" addressed most concerns of both reviewers and has in my opinion substantially improved. The manuscript explains now in detail why the authors focus on Treg and Th17 cells context of Crohn's disease. The addition of Th1 and Th2 data helps to better understand the main impact of CD39 and CD39-AS on Treg and Th17 cells. The discussion of raised points concerning the contribution of Treg and Th17 cells to the mucosal inflammation is thoroughly elaborated and necessary additions have been made. The utilization of polarized human T cells along with cell-lines provides a better insight into how the Treg population suppresses inflammation in the presence of FANA-CD39-AS. of Treg and Th17. The implementation of new experiments, respective controls, and increase of the sample size on key experiments served very well to improve the overall quality of the manuscript.

There are only a few points that I would like to mention.

The gating strategy figure lacks dead cells and duplet exclusion steps. Demonstration of each gating step will be a complete guide for researchers who aim to reproduce the conditions and will help readers to follow cellular identification steps. Moreover, there seems to be a technical issue in the analysis of the flow cytometry data in supplementary figure 8. For each tissue there is a different set of quadrants, in some cases, cell populations are "cut" with gating lines, adding isotype control staining to the figure may help to understand the different cut-off levels in different tissues. CD4 positivity is also a concern, ranging between 30-40%, for example in supl.fig.8, panel D: all of the cells seem to be IL-17 positive, it is not clear which type of cells are these CD4- IL-17+ cells.

Re: NCOMMS-20-02176B

Point by point reply to Reviewers

Reviewer # 2

The revised manuscript by Harshe and colleagues entitled “Endogenous antisense RNA curbs CD39 expression in Crohn’s disease” addressed most concerns of both reviewers and has in my opinion substantially improved. The manuscript explains now in detail why the authors focus on Treg and Th17 cells context of Crohn’s disease. The addition of Th1 and Th2 data helps to better understand the main impact of CD39 and CD39-AS on Treg and Th17 cells. The discussion of raised points concerning the contribution of Treg and Th17 cells to the mucosal inflammation is thoroughly elaborated and necessary additions have been made. The utilization of polarized human T cells along with cell-lines provides a better insight into how the Treg population suppresses inflammation in the presence of FANA-CD39-AS. of Treg and Th17. The implementation of new experiments, respective controls, and increase of the sample size on key experiments served very well to improve the overall quality of the manuscript.

Author’s reply: We thank the Reviewer for the positive feedback.

1. There are only a few points that I would like to mention.

Author’s reply: Please see below our answer to your comments.

2. The gating strategy figure lacks dead cells and duplet exclusion steps. Demonstration of each gating step will be a complete guide for researchers who aim to reproduce the conditions and will help readers to follow cellular identification steps.

Author’s reply: We thank the Reviewer for the observation and comment. We have now shown the gating strategy and included dead cell and doublet exclusion steps in newly added flow cytometry plots in Supplementary Figure 8A. Dead cell and doublet exclusion steps are now provided for the flow cytometry analysis of lymphocytes obtained from peripheral blood, spleen, mesenteric lymph node, intraepithelial and lamina propria compartments (Supplementary Figure 8A). The figure legend and text in the methodology section have been updated accordingly.

Moreover, there seems to be a technical issue in the analysis of the flow cytometry data in supplementary figure 8. For each tissue there is a different set of quadrants, in some cases, cell populations are “cut” with gating lines, adding isotype control staining to the figure may help to understand the different cut-off levels in different tissues. CD4 positivity is also a concern, ranging between 30-40%, for example in supl.fig.8, panel D: all of the cells seem to be IL-17 positive, it is not clear which type of cells are these CD4- IL-17+ cells.

Author’s reply: We thank the Reviewer for pointing this out. The analysis of CD4+CD39+, CD4+FOXP3+, CD4+IL17+ and CD4+IL10+ cells has now been performed again and new representative flow cytometry plots from vehicle and FANA-CD39-AS oligonucleotide treated mice have been included in Supplementary Figure 8B-D. As recommended, relevant isotype

control staining is now shown for each subpopulation in all compartments (Supplementary Figure 8B-D). The Figure legend and text in the methodology section have been updated accordingly. The flow cytometry plots shown in Figure 4F have been also updated.

Please note that, after re-running the analysis, the frequency of non-CD4 cells positive for CD39, FOXP3, IL17 or IL10 (possibly representing non-specific staining due to cell debris derived from non-lymphoid cell populations) has decreased substantially while the proportion of total CD4 cells is consistently higher than the 10% re-population threshold in all samples.

REVIEWERS' COMMENTS

Reviewer #2 (Remarks to the Author):

I do not have any questions anymore.

REVIEWERS' COMMENTS

Reviewer #2 (Remarks to the Author):

I do not have any questions anymore.

Author's reply: We thank the Reviewer for the feedback